# The emergence and evolution of gene expression in genome regions replete with regulatory motifs

**Timothy Fuqua[1,2], Yiqiao Sun[1,2], Andreas Wagner[1,2,3]***

[1]Department of Evolutionary Biology and Environmental Studies, University of Zurich, Zurich, Switzerland; [2]Swiss Institute of Bioinformatics, Quartier Sorge-Batiment Genopode, Lausanne, Switzerland; [3]The Santa Fe Institute, Santa Fe, United States

## eLife Assessment

This **important** study explores the relationship between the sequence of prokaryotic promoter elements and their activity using mutagenesis to generate thousands of mutant sequences. The evidence supporting these findings is **convincing**. This work will appeal to those interested in bacterial genetics, genome evolution, and gene regulation.

***For correspondence:**
andreas.wagner@ieu.uzh.ch

**Competing interest:** The authors declare that no competing interests exist.

**Abstract** Gene regulation is essential for life and controlled by regulatory DNA. Mutations can modify the activity of regulatory DNA, and also create new regulatory DNA, a process called regulatory emergence. Non-regulatory and regulatory DNA contain motifs to which transcription factors may bind. In prokaryotes, gene expression requires a stretch of DNA called a promoter, which contains two motifs called −10 and −35 boxes. However, these motifs may occur in both promoters and non-promoter DNA in multiple copies. They have been implicated in some studies to improve promoter activity, and in others to repress it. Here, we ask whether the presence of such motifs in different genetic sequences influences promoter evolution and emergence. To understand whether and how promoter motifs influence promoter emergence and evolution, we start from 50 'promoter islands', DNA sequences enriched with −10 and −35 boxes. We mutagenize these starting 'parent' sequences, and measure gene expression driven by 240,000 of the resulting mutants. We find that the probability that mutations create an active promoter varies more than 200-fold, and is not correlated with the number of promoter motifs. For parent sequences without promoter activity, mutations created over 1500 new −10 and −35 boxes at unique positions in the library, but only ~0.3% of these resulted in de-novo promoter activity. Only ~13% of all −10 and −35 boxes contribute to de-novo promoter activity. For parent sequences with promoter activity, mutations created new −10 and −35 boxes in 11 specific positions that partially overlap with preexisting ones to modulate expression. We also find that −10 and −35 boxes do not repress promoter activity. Overall, our work demonstrates how promoter motifs influence promoter emergence and evolution. It has implications for predicting and understanding regulatory evolution, de novo genes, and phenotypic evolution.

## Introduction

Gene regulation is critical to the development and homeostasis of living beings (**Prud'homme et al., 2007**). It is controlled by DNA sequences that encode motifs for transcription factors (TFs) to bind and either activate or repress transcription (**Jacob and Monod, 1961**; **Seshasayee et al., 2011**). Mutations to these motifs can modulate gene expression, leading to the evolution of new phenotypes

(*Fuqua et al., 2020*; *Prud'homme et al., 2007*). Mutations can also create new regulatory sequences from non-regulatory DNA, a process called *regulatory emergence* (*Rebeiz et al., 2011*; *Villar et al., 2015*).

A regulatory motif is a DNA sequence computationally predicted to be compatible with TF binding. Although it is commonly thought that regulatory motifs influence both regulatory emergence and evolution, pertinent empirical evidence is sparse and conflicting. On the one hand, such evidence shows that motifs can facilitate regulatory emergence and evolution. For example, recent experiments showed that a synthetic random DNA molecule with a single motif for a particular TF favored the emergence of tissue-specific gene expression for that TF (*Galupa et al., 2023*). Likewise, transposable elements contain motifs compatible with the binding of particular TFs (*Lynch et al., 2015*; *Villanueva-Cañas et al., 2019*), and these motifs have biased gene regulatory networks towards utilizing these TFs in evolution (*Lynch et al., 2015*). In *E. coli*, an active promoter is more likely to overlap with multiple promoter motifs than individual motifs (*Huerta and Collado-Vides, 2003*).

On the other hand, there is also evidence that regulatory motifs can impede regulatory emergence and evolution. One reason is that TFs can compete in their binding to shared motifs (*Chauhan et al., 2022*; *Lagator et al., 2022*). For example, the TF Histone-like nucleoid-structuring protein (H-NS) binds to sequences similar to prokaryotic promoters to repress spurious transcription and the expression of foreign DNA (*Forrest et al., 2022*; *Oshima et al., 2006*). Another reason is that gene expression can be silenced when regulatory DNA contains multiple motifs for the same TF (*Berman et al., 2002*; *Bykov et al., 2020*; *Loell et al., 2024*; *Perales et al., 2016*). For example, in mice, retinal 'silencers' contain clusters of motifs for the TF CRX, while retinal 'enhancers' contain a mixture of CRX and other TF motifs (*Loell et al., 2024*). A third reason is that clusters of motifs for RNA polymerase binding can cause collisions between RNA polymerase molecules, leading to premature termination of transcription (*Crampton et al., 2006*; *Hobson et al., 2012*; *Wang et al., 2023*).

A canonical prokaryotic promoter recruits the RNA polymerase holoenzyme with its subunit σ70 to transcribe downstream sequences (*Burgess et al., 1969*; *Huerta and Collado-Vides, 2003*; *Paget and Helmann, 2003*; *van Hijum et al., 2009*). The 'standard model' (*Lagator et al., 2022*) of such promoters is that σ70 binds to two motifs: the −10 box, and the −35 box (*Huerta and Collado-Vides, 2003*; *Pribnow, 1975*; *Urtecho et al., 2019*). These motifs are thoroughly characterized (consensus TATAAT and TTGACA, respectively) and are optimally spaced 17±1 bp apart, with transcription beginning 7 bp downstream of the −10 box (*Belliveau et al., 2018*; *Ireland et al., 2020*; *Kinney et al., 2010*; *Urtecho et al., 2023*; *Urtecho et al., 2019*).

This standard model, however, is an oversimplification of promoter architecture (*Lagator et al., 2022*). Studies have also identified an AT-rich motif upstream of the −35 box called the UP element (consensus: AAAWWTWTTTTnnAAAA) (*Estrem et al., 1998*; *Ross et al., 1993*), which can increase promoter activity (*Bracco et al., 1989*; *Meng et al., 2001*). However, only ~3–19% of all characterized promoters have an UP-element (*Estrem et al., 1998*). Recent work also identified an AT-rich motif within the spacer region associated with promoter activity (*Warman et al., 2020*). Other characterized promoter motifs include variations of the −10 box such as the *extended −10 box / TGn motif* (TGnTATAAT), which can sometimes compensate for an absent −35 box (*Burr et al., 2000*; *Mitchell et al., 2003*; *Yona et al., 2018*). A symmetrical −10 box variant also leads to bidirectional promoter activity (consensus: TATWATA) if correctly spaced between two flanking −35 boxes located on opposing strands (*Warman et al., 2021*). Like promoters in humans (*Gotea et al., 2010*) and invertebrates (*Lifanov et al., 2003*), most prokaryotic promoters have multiple regulatory motifs (−10 and −35 boxes; *Gama-Castro et al., 2016*; *Huerta and Collado-Vides, 2003*) and transcription start sites (*Mendoza-Vargas et al., 2009*). It is not clear whether these additional motifs influence promoter activity. If so, it may help to explain why it is difficult to predict promoter strength and activity (*Lagator et al., 2022*; *Urtecho et al., 2023*).

Recent work shows that mutations can help new promoters to emerge from promoter motifs or from sequences adjacent to such motifs (*Bykov et al., 2020*; *Fuqua and Wagner, 2023*; *Yona et al., 2018*). However, encoding −10 and −35 boxes is insufficient to drive complete transcription of a gene coding sequence. For instance, the *E. coli* genome contains clusters of −10 and −35 boxes that are bound by RNA polymerase and produce short oligonucleotide fragments, but rarely create complete transcripts. Such clusters are called *promoter islands*, and are strongly associated with horizontally

transferred DNA (*Bykov et al., 2020*; *Panyukov and Ozoline, 2013*; *Purtov et al., 2014*; *Shavkunov et al., 2009*).

There are two proposed explanations for why promoter islands do not create full transcripts. First, the TF H-NS may repress promoter activity in promoter islands. This is because in a *Δhns* background, transcript levels from the promoter islands increases (*Purtov et al., 2014*). However, mutagenizing a specific promoter island (*appY*) until it transcribes a GFP reporter, reveals that in vitro H-NS binding does not significantly change when GFP levels increase (*Bykov et al., 2020*). Thus, it is not clear whether H-NS actually represses the complete transcription of these sequences. The second proposed explanation is that excessive promoter motifs silence transcription. The aforementioned study found that promoter activity increases when mutations improve a –10 box to better match its consensus (TAAAAA→TAtAct), while simultaneously destroying surrounding –10 and –35 boxes (*Bykov et al., 2020*). However, we note that if these surrounding motifs never contributed to GFP fluorescence to begin with, then mutations could also simply have accumulated in them during random mutagenesis without affecting promoter activity.

In this study, we define a promoter as a DNA sequence that drives the expression of a (fluorescent) protein whose expression level, measured by its fluorescence, is greater than a defined threshold. We use a threshold of 1.5 arbitrary units (a.u.) of fluorescence. This definition does not distinguish between transcription and translation. We chose it because protein expression is usually more important than RNA expression whenever natural selection acts on gene expression, because it is the primary pheno-type visible to natural selection (*Jiang et al., 2023*).

We systematically study the ability of promoter motifs to influence the evolution and emergence of promoter activity. To this end, we select 25 150 bp *template sequences* with and without promoter activity (*P1-P25*), from 25 independent promoter islands. For each template, we treated both strands as independent *parent sequences* (N=50 parent sequences). We then screened over 240,000 mutant variants (*daughter sequences*) derived from these parent sequences for promoter activity. In non-promoter parent sequences, the emergence of new –10 and –35 boxes in daughter sequences rarely resulted in de novo promoter activity. Conversely, for parent sequences with existing promoter activity, the promoter function was only modulated when new –10 and –35 boxes formed in daughter sequences that partially overlapped with existing boxes. Notably, we found no evidence that –10 and –35 boxes directly repress promoter activity, although H-NS does repress some of the promoter islands.

## Results

### Promoter islands are enriched with motifs for -10 and -35 boxes

Promoter islands are ~1 kbps (kilo base pairs) long (*Bykov et al., 2020*; *Panyukov and Ozoline, 2013*; *Purtov et al., 2014*; *Shavkunov et al., 2009*), but for adequate mutational coverage, it is better to work with shorter sequences. We thus identified within 25 promoter islands a region of 150 bp that showed the highest density of predicted –10 and –35 boxes for further analysis (*Figure 1A*). To this end, we used position-weight matrices (PWMs) – computational tools to identify these promoter motifs (Methods; *Hertz and Stormo, 1999*). These PWMs are derived from 145 of the top predicted –10 and –35 boxes from the database RegulonDB (*Tierrafría et al., 2022*), and are highly correlated with experimentally-derived –10 and –35 boxes (*Belliveau et al., 2018*; see Methods). There are 25 *template sequences* (P1-25), and each genetic strand is an independent *parent sequence* (N=50 parents). On average, each parent sequence contains ~5.32 –10 boxes and ~7.04 –35 boxes (*Figure 1— figure supplement 1*). 18 of these –10 boxes also include the TGn motif upstream of the hexamer.

### Non-promoters vary widely in their potential to become promoters

Next we created a mutant library of *daughter* sequences from the 25 templates using an error-prone polymerase chain reaction (*Figure 1B*), and measured the extent to which these daughter sequences can drive gene expression using the well-established sort-seq procedure (*Kinney et al., 2010*; *Peterman and Levine, 2016*). Specifically, we cloned the library into a bidirectional fluorescence reporter plasmid (pMR1) to measure promoter activity on both DNA strands (top: GFP, bottom: RFP; *Guazzaroni and Silva-Rocha, 2014*; *Westmann et al., 2018*). We then transformed the plasmid library into *E. coli* (DH5α, Takara Japan), sorted the bacteria using fluorescence activated cell sorting

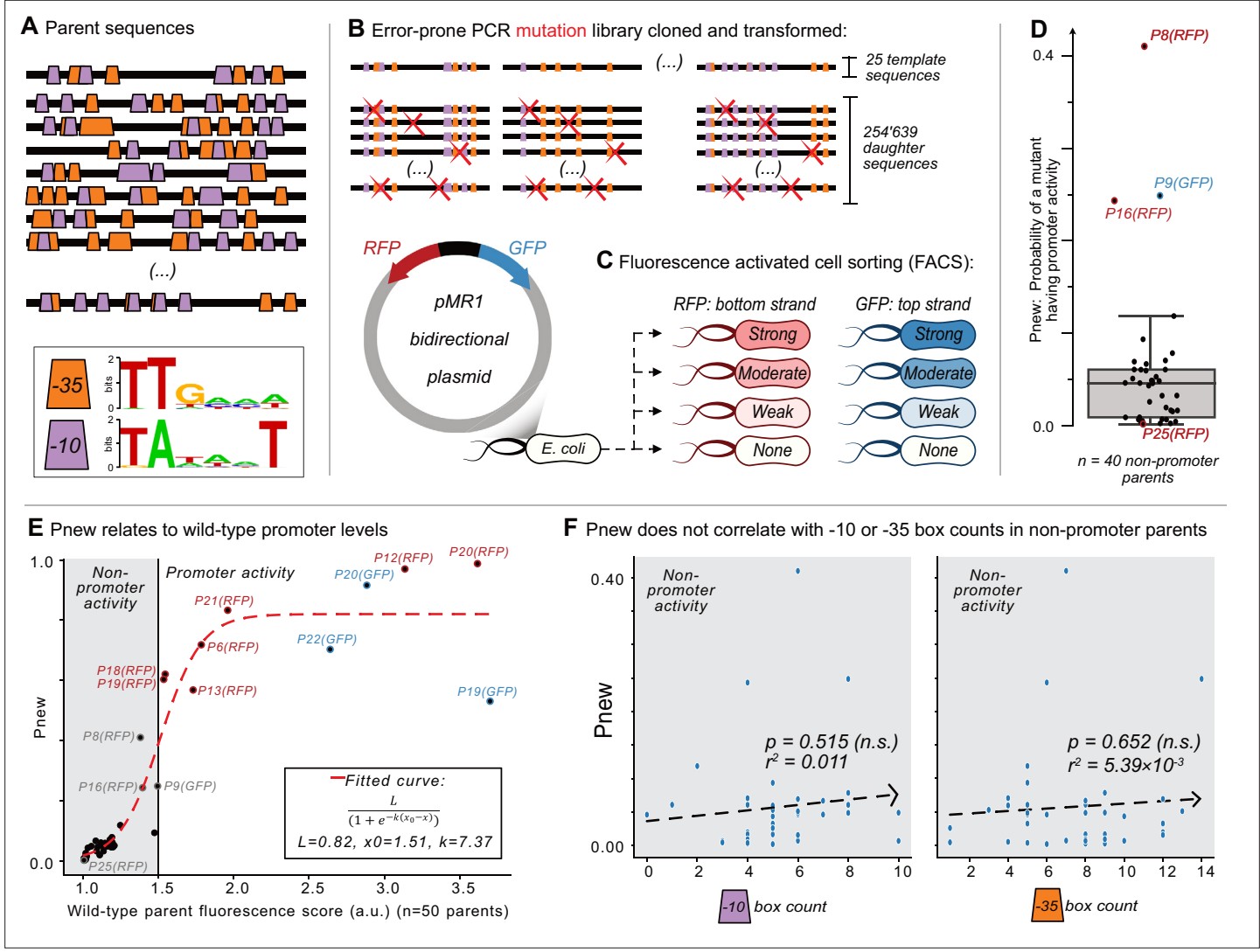

**Figure 1.** Mutagenesis reveals vastly different probabilities of promoter emergence. (**A**) The location of –10 and –35 boxes in a subset of the parent sequences. See *Figure 1—figure supplement 1* for the complete set (n=50). Orange trapezoids correspond to –35 boxes, and magenta trapezoids to –10 boxes, each identified using position-weight matrices (see sequence logos below and *Figure 1—figure supplement 1*). (**B–C**) The Sort-Seq protocol. (**B**) Top: we amplified 25 150 bp template sequences from 25 promoter islands in the *E. coli* genome with an error-prone polymerase chain reaction, generating a mutagenesis library of 245,639 unique daughter sequences. Bottom: we cloned the library into the pMR1 reporter plasmid between a green fluorescent protein (GFP) coding sequence on the top strand (blue arrow) and a red fluorescent protein (RFP) coding sequence on the bottom strand (red arrow). We transformed the plasmid library into *E. coli*. (**C**) Using fluorescence-activated cell sorting (FACS), we sorted the transformed *E. coli* cells into 8 fluorescence bins: none, weak, moderate, and strong, for both RFP and GFP expression (see *Figure 1—figure supplement 2* for bins). We sequenced the plasmid inserts of cells from each bin, assigning a fluorescence score from 1.0 to 4.0 arbitrary units (a.u.), ranging from no fluorescence (1.0 a.u.) to strong fluorescence (4.0 a.u.) (see *Figure 1—figure supplement 4* for score distributions). (**D**) Pnew is the ratio of daughter sequences with a fluorescence score greater than 1.5 a.u. to the total number of daughter sequences for each parent. Circles show the Pnew values for each of 40 non-promoter parents (i.e. 50 parents – 3 top strand promoters – 7 bottom strand promoters = 40 non-promoter parents.) The height of the box represents the interquartile range (IQR) and the center line the median. Whiskers correspond to 1.5×IQR. (**E**) A scatterplot between the calculated fluorescence score (Methods) for all of the parent sequences (n=50, both with and without promoter activity) and Pnew. The solid horizontal line at 1.5 a.u. marks our cutoff for whether a parent does not (gray shaded region) or does have promoter activity (no shading). The dashed line corresponds to a fitted sigmoidal curve, where L is equal to the upper asymptote, x0 is the inflection point, and k is the slope. See Methods for fitting details. (**F**) Scatterplots comparing the number of –10 or –35 box counts per parent sequence and their respective Pnew values. The dashed line is the line of best fit calculated using the method of least squares. We test the null hypothesis that the slope of the correlation equals zero with the Wald Test. The $r^2$ value is the Pearson correlation coefficient. Calculation carried out using scipy.stats.linregress (version 1.8.1). Left: the number of –10 box counts (p=0.515, $r^2$=0.011). Right: the number of –35 box counts (p=0.652, $r^2$=5.39 × $10^{-3}$).

The online version of this article includes the following figure supplement(s) for figure 1:

*Figure 1 continued on next page*

*Figure 1 continued*

**Figure supplement 1.** Promoter island sequences.

**Figure supplement 2.** Mutagenesis library and sort-seq bins.

**Figure supplement 3.** Mutational coverage for the template sequences.

**Figure supplement 4.** Histogram of the fluorescence score distributions for each parent and its respective daughter sequences.

**Figure supplement 5.** Correlation of $P_{new}$ with sequence composition.

**Figure supplement 6.** Molecular cloning of parent sequences.

(FACS) into eight fluorescence bins (none, weak, moderate, and strong fluorescence, for both GFP and RFP) (*Figure 1C*, see also *Figure 1—figure supplement 2*), and sequenced the inserts from the sorted fluorescence bins (see Methods).

Our library comprised 245,639 unique daughter sequences, with an average of 4'913 daughter sequences per template and ~2.5-point mutations per daughter (*Figure 1—figure supplement 2*). Each template has 3×150 possible mutational neighbors that differ by a single mutation. We define coverage as the percentage of these 450 bases and positions that are present in at least one daughter sequence for each parent. (Daughters contain 1–10 point mutations). The mean coverage was 93% (minimum: 80%; maximum: 100%; *Figure 1—figure supplement 3*). The mutation library therefore encompasses the majority of neighboring mutations for the 25 templates.

We next quantified how strongly each daughter sequence could drive gene expression from the top and bottom strand by calculating a fluorescence score for both GFP and RFP expression, respectively. These scores are bounded between a lowest score of 1.0 arbitrary units (a.u.), indicating no expression, and a highest score of 4.0 a.u., indicating maximal expression (methods). We declared a daughter sequence to have *promoter activity* or to be a *promoter* if its score was greater than or equal to 1.5 a.u., as this score lies at the boundary between no fluorescence and weak fluorescence based on the sort-seq bins (Methods). Otherwise, we refer to a daughter sequence as having *no promoter activity* or being a *non-promoter*.

Because some sequences in our library are unmutated parent sequences, we determined that 10/50 of the parent sequences already encode promoter activity before mutagenesis. Specifically, three parents drove expression on the top strand (P19-GFP, P20-GFP, P22-GFP), and seven did on the bottom strand (P6-RFP, P12-RFP, P13-RFP, P18-RFP, P19-RFP, P20-RFP, P21-RFP). Two parents harbor bidirectional promoters (P19 and P20). The remaining 40 parent sequences are non-promoters, with an average fluorescence score of 1.39 a.u. We note that some of these parents have a fluorescence score higher than 1.39 a.u., but less than 1.50 a.u. such as P8-RFP (1.38 a.u.), P16-RFP (1.39 a.u.), P9-GFP (1.49 a.u.), and P1-GFP (1.47 a.u.). Whether these are truly 'promoters' or not, is based solely on our threshold value of 1.5 a.u. We also note that 30% (15/50) of parents have the TGn motif upstream of a –10 box, but only 20% (3/15) of these parents have promoter activity (underlined with promoter activity: P4-RFP, P6-RFP, P8-RFP, P9-RFP, P10-RFP, P11-GFP, P12-GFP, P17-GFP, P18-GFP, P18-RFP, P19-RFP, P22-RFP, P24-GFP, P25-GFP, P25-RFP). See *Figure 1—figure supplement 4* for fluorescence score distributions for each parent and its daughters, and *Source data 2* for all daughter sequence fluorescence scores.

## Promoter emergence correlates with minute differences in background promoter levels

Although mutating each of the 40 non-promoter parent sequences could create promoter activity, the likelihood $P_{new}$ that a mutant has promoter activity, varies dramatically among parents. For each non-promoter parent, *Figure 1D* shows the percentage of active daughter sequences. The median $P_{new}$ is 0.046 (std. ±0.078), meaning that ~4.6% of all mutants have promoter activity. The lowest $P_{new}$ is 0.002 (P25-GFP) and the highest 0.41 (P8-RFP), a 205-fold difference.

We hypothesized that these large differences in $P_{new}$ could be explained by minute differences in the fluorescence scores of each parent, particularly if its score was below 1.5 a.u. Plotting the fluorescence scores of each parent (N=50) and their respective $P_{new}$ values as a scatterplot (*Figure 1E*), we can fit these values to a sigmoid curve (see methods). This finding helps to explain why P8-RFP has a high $P_{new}$ (0.41) and P25-GFP a low $P_{new}$ (0.002), as their fluorescence scores are 1.380 and 1.009 a.u.,

respectively. The fact that the inflection point of the fitted curve is at 1.51 a.u. further justifies our use of 1.5 a.u. as a cutoff for promoter and non-promoter activity.

## Promoter emergence does not correlate with simple sequence features

We asked what else could explain the differences in $P_{new}$ values for the non-promoter parents. To this end, we calculated linear regressions between $P_{new}$ values of the non-promoter parents and various sequence features. These include the number of daughter sequences, GC-content, –10 box counts and –35 box counts. We found that $P_{new}$ does not significantly correlate with any of these features, including the number of –10 or –35 boxes (least squares linear regression, p=0.529, 0.930, 0.515, 0.652, respectively; *Figure 1F*, *Figure 1—figure supplement 5A, B*).

We then asked whether any k-mers ranging from 1 to 6 bp correlated with the non-promoter $P_{new}$ values (5,460 possible k-mers). 718 of these 1–6 bp k-mers are present 3 or more times in at least one non-promoter parent. We calculated a linear regression between the frequency of these 718 k-mers and each $P_{new}$ value, and adjusted the p-values to respective q-values (Benjamini-Hochberg correction, FDR = 0.05). This analysis revealed six k-mers: CTTC, GTTG, ACTTC, GTTGA, AACTTC, TAACTT which correlate with $P_{new}$. Five of the six k-mers contain either GTTG or CTTC (underlined). However, these correlations are heavily influenced by an outlying $P_{new}$ value of 0.41 (P8-RFP; *Figure 1—figure supplement 5C–H*), and upon removing P8-RFP from the analysis, no k-mer significantly correlates with $P_{new}$ (data not shown).

## Promoters emerge and evolve only from specific subsets of -10 and -35 boxes

Next, we asked where both existing and new promoters are located within the parent sequences. To this end, we first calculated for each parent and its respective daughters, the mutual information $I_i\left(b,f\right)$ between the nucleotide identity at each position $i$ and the corresponding fluorescence level $f$ (*Figure 2A*). The essence of this calculation is to determine for every position $i$, the probability $p_i\left(b\right)$ of that position being an A,T,C, or G and the probability of each sequence having a (rounded) fluorescence score of 1,2,3, or 4 a.u. $p\left(f\right)$. Representing these probabilities as circles in a Venn-diagram, the overlap between the circles is the joint probability $p_i\left(b,f\right)$ that position $i$ has a specific base $b$ AND a particular fluorescence score $f$. The larger this joint probability is compared to the individual probabilities, the more important the nucleotide identity at position $i$ is for fluorescence, that is for promoter activity. This calculation can reveal positions where RNA polymerase and transcription factors bind (*Belliveau et al., 2018*; *Ireland et al., 2020*; *Kinney et al., 2010*), as well as DNA regions where promoters most readily emerge (*Fuqua and Wagner, 2023*). The calculation effectively converts all 245,639 data points into maps that reveal where promoters either already exist or emerge in each of the parent sequences (see Methods).

We identified 'hotspots' – regions of especially high mutual information – for each parent sequence (*Fuqua and Wagner, 2023*) (see methods and *Source data 3*). For example, two such hotspots exist in parent P19-GFP, which is an active promoter that drives expression from both DNA strands (*Figure 2B*). One hotspot overlaps with a –35 box and the other overlaps with a –10 box. These hotspots indicate that fluorescence changes when these DNA sequences mutate. Because these sequences additionally match a canonical promoter, they are the likely cause of the promoter activity in P19-GFP (see *Figure 4—figure supplement 2A–A'* for further validation).

Across all 50 parent sequences, we identified a total of 68 hotspots (*Figure 2C*). 34 hotspots occur in the parent sequences with promoter activity. They correspond to locations where mutations either increase or decrease promoter activity. ~15% (5/34) of these hotspots overlap exclusively with –10 boxes, ~32% (11/34) with –35 boxes, and ~18% (6/34) overlap with both –10 and –35 boxes. The remaining ~35% of hotspots overlap with neither a –10 nor a –35 box (12/34). To find out whether these overlaps could be explained by chance alone, we computationally scrambled the parent sequences while maintaining the hotspot coordinates. ~9% (3/34) of the hotspots overlapped with –10 boxes in the scrambled parents, ~12% (4/34) with –35 boxes, ~3% (1/34) with both –10 and –35 boxes, and the remaining ~76% (26/34) overlapping with neither. Hotspots in the biological parent sequences overlap to a significantly greater extent with the two classes of boxes than the scrambled parent sequences (chi-squared test, 3 degrees of freedom, $\chi^2$=46.1, p=5.34 × 10$^{-10}$). This analysis shows that the overlap does not simply result from the high density of –10 and –35 boxes.

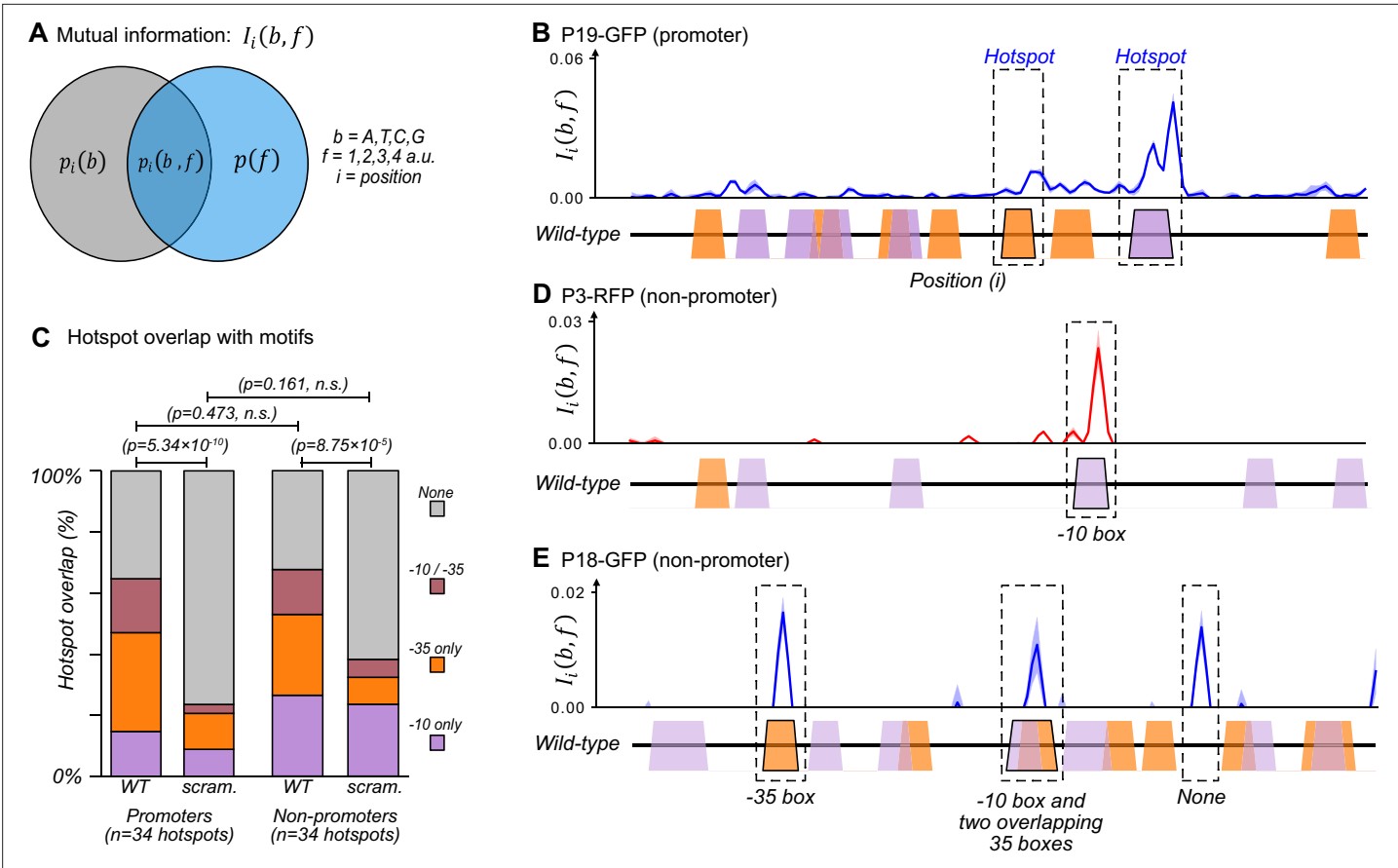

**Figure 2.** The majority of promoters emerge and evolve within a subset of preexisting promoter motifs. (**A**) We calculated the mutual information $I_i(b,f)$ between nucleotide identity ($b$=A,T,C,G) and fluorescence scores rounded to the nearest whole number ($f$=1,2,3,4 a.u.) for each position $i$ in a parent sequence. In essence, the calculation compares the probability $p_i(b)$ of a base $b$ occurring at position $i$, and the probability $p(f)$ that a sequence has a fluorescence score $f$. The joint probability $p_i(b,f)$ is the probability that a sequence with base $b$ at position $i$ has fluorescence score $f$. The greater the joint probability is compared to the individual probabilities, the more important the base at this position is for promoter activity. See methods. (**B**) An example of how to interpret mutual information using position-weight matrix (PWM) scores of predicted −10 and −35 boxes. Top: we plot the mutual information $I_i(b,f)$ for P19-GFP. P19-GFP is an active promoter. Solid line: mean mutual information. Shaded region:±1 standard deviation when the dataset is randomly split into three equally sized subsets (Methods). Bottom: position-weight matrix (PWM) predictions for the −10 boxes (magenta trapezoids) and −35 boxes (orange trapezoids) along the wild-type parent sequence. We define hotspots as mutual information peaks greater than or equal to the 90th percentile of total mutual information (Methods), and highlight them with dashed rectangles. (**C**) Stacked bar plots depicting the percentage of hotspots overlapping with −10 boxes only (magenta), −35 boxes (orange), both −10 and −35 boxes (red), or with neither (gray). We plot this information for the wild-type (WT) promoters and non-promoter parents, as well as scrambled (scram.) versions of the parents. Horizontal lines correspond to $\chi$ (chi-squared) tests between the counts in each group (3 degrees of freedom). (**D**) Analogous to (**B**) but for parent P3-RFP. P3-RFP is an inactive parent sequence. Hotspot overlaps with a −10 box. (**E**) Analogous to (**B**) but for parent P18-GFP. P18-GFP is an inactive parent sequence. Hotspots overlap with (from left to right) a −35 box, both a −10 and a −35 box, and neither (None). *Figure 2—figure supplement 1* shows analogous mutual information plots for daughters derived from each parent sequence.

The online version of this article includes the following figure supplement(s) for figure 2:

**Figure supplement 1.** Mutual information and promoter motifs in the parent sequences.

The remaining 34 hotspots occur in non-promoter parent sequences. Mutations in these locations solely increase fluorescence. Of these 34 hotspots, ~26% (9/34) overlap with −10 boxes, ~26% (9/34) with −35 boxes, ~15% with both −10 and −35 boxes (5/34), and ~32% with neither motif (11/34) (*Figure 2C–E*). Scrambling the parent sequences while maintaining hotspot coordinates resulted in ~24% (8/34) of hotspots overlapping with a −10 box, ~9% (3/34) with a −35 box, ~6% (2/34) with both −10 and −35 box, and the remaining ~62% (21/34) with neither. Again, these percentages of overlap with existing boxes are significantly greater for biological than for scrambled parent sequences (chi-squared test, 3 degrees of freedom, $\chi^2$=21.4, p=8.75 × 10⁻⁵). There is no significant difference

between the motif overlap of promoters and non-promoters (chi-squared test, 3 degrees of freedom, $\chi^2$=2.51, p=0.473). See *Figure 2—figure supplement 1* for mutual information plots for all parent sequences and *Source data 3* for a table with the hotspots and their coordinates.

Despite this extensive overlap between promoter motifs and hotspots, each parent contains additional –10 and –35 boxes that do not overlap with hotspots. Of the 266 –10 boxes, only 37 (~14%) overlap within ±3 bp from a hotspot. Similarly, of the 352 –35 boxes, only 43 (~12%) overlap with a hotspot. For example, the non-promoter parent P3-RFP contains five –10 boxes, but only one overlaps with a hotspot (*Figure 2D*). Similarly, the non-promoter P18-GFP contains ten –10 and nine –35 boxes, but promoters only emerge in three hotspots of P18-GFP (*Figure 2E*).

In sum, while the majority (~67%) of new promoters evolve and emerge from –10 and –35 boxes, the presence of one such box does not indicate that promoter activity can emerge from it, or is encoded within it. Only ~13% of all –10 and –35 boxes in the parents contribute to promoter activity.

## New -10 and -35 boxes readily emerge, but rarely lead to de-novo promoter activity

We hypothesized that promoters emerge when mutations create new –10 and –35 boxes. To test this hypothesis, we examined each hotspot and used PWMs to find out whether a subset of the daughter sequences gains a –10 or –35 box in the hotspot. We then asked whether gaining a –10 or –35 box leads to a significant increase in fluorescence that indicates the creation of a new promoter (Mann-Whitney U-test, *Figure 3—figure supplement 1* and methods). See *Source data 4* for the coordinates, fluorescence changes, and significance for the –10 and –35 boxes.

Mutations indeed created many new –10 and –35 boxes in our daughter sequences. On average, 39.5 and 39.4 new –10 and –35 boxes emerged at unique positions within the daughter sequences of each mutagenized parent (*Figure 3A and B*), with 1'562 and 1'576 new locations for –10 boxes and –35 boxes, respectively. ~22% (684/3138) of these new boxes are spaced 15–20 bp away from their cognate box, and ~7.3% (114/1562) of the new –10 boxes have the TGn motif upstream of them. However, only a mere five of the new –10 boxes and four of the new –35 boxes are significantly associated with increasing fluorescence by more than +0.5 a.u. (*Figure 3C and D*). That is, only ~0.3% of all new –10 and –35 boxes actually create new promoter activity. Thus, the creation of a –10 or –35 box, even in the appropriate proximity to the cognate box, rarely leads to de novo promoter activity.

## Promoters can emerge when mutations create motifs but not by destroying them

We briefly highlight two examples where gaining –10 and –35 boxes leads to de-novo promoter activity. Parent P16-RFP gains promoter activity when mutations create a –10 box 17 bp downstream of a preexisting –35 box (*Figure 3E*), increasing median fluorescence by +1.99 a.u. (*Figure 3F*; two-tailed MWU test, q=7.64 × 10$^{-32}$). The majority of these –10 boxes emerge when the sequence TAAACA (0.44 bits) mutates to TAAACt (7.28 bits). To validate this observation experimentally, we created this point mutation in a P16-RFP reporter, and compared its fluorescence levels with the wild-type P16-RFP using a plate reader (see Methods). This experiment shows that fluorescence increases ~11.2 fold when the –10 box emerges (*Figure 3F'*).

We also highlight one hotspot where gaining a –35 box creates de novo promoter activity. Specifically, parent P1-RFP gains promoter activity when mutations create a –35 box 18 bp upstream of a preexisting –10 box (*Figure 3G*), which increases fluorescence by +0.74 a.u. (*Figure 3H*, two-tailed MWU test, q=4.25 × 10$^{-42}$). The majority of new –35 boxes emerge when the sequence ATGAGT (0.00 bits) mutates to tTGAGT (3.93 bits). A validation construct with this point mutation shows a ~2.0-fold increase in fluorescence compared to the wild-type (*Figure 3H'*).

*Figure 3—figure supplement 2* shows additional examples in which –35 and –10 boxes are created, and the results of their validation experiments. In two of these hotspots, our validation experiments revealed no substantial difference in gene expression as a result of the hotspot mutation (*Figure 3—figure supplement 2F', J'*). In both of these false positives, new –10 boxes emerge in locations without an upstream –35 box.

To summarize, mutations readily create new –10 boxes and –35 boxes (see *Figure 3A–B*). This can create de-novo promoter activity (see *Figure 3E-H*, *Figure 3—figure supplement 2*), but it rarely

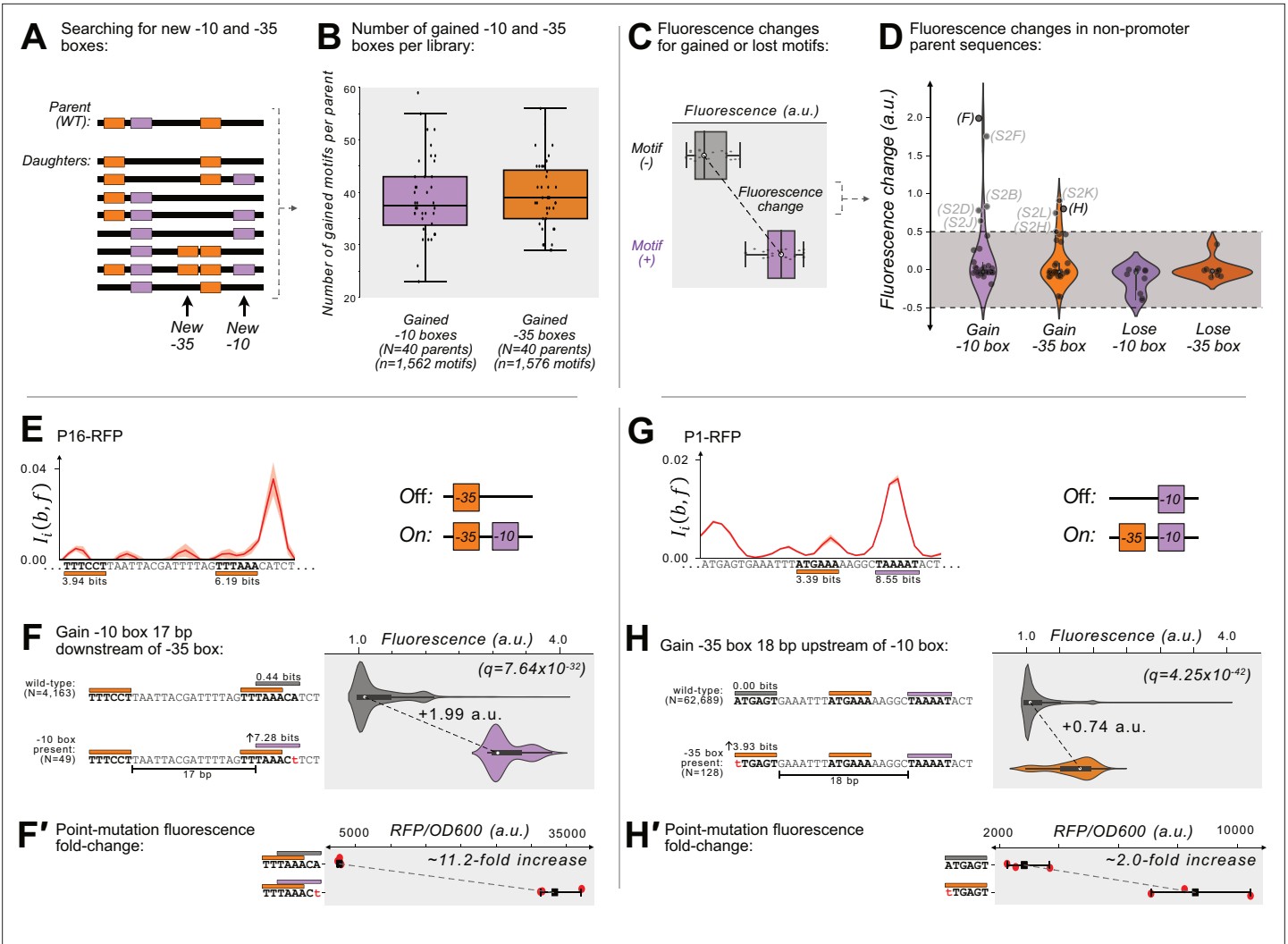

**Figure 3.** Gaining −10 and −35 boxes rarely creates de novo promoters. (**A**) A cartoon for how we identify new −35 and −10 boxes (orange and magenta boxes, respectively) within each DNA sequence (thick, black, horizontal bars). Top: we first identify the −10 and −35 boxes in the wild-type parent sequence. Bottom: within all of the daughter sequences, we identify all locations where new −10 and −35 boxes appear, as shown with the arrows below the daughter sequences. We then count the new motifs gained for each parent, and plot the results in a box plot (**B**) for both new −10 and −35 boxes. Boxes represent the interquartile range (IQR) and the center line the median. Whiskers correspond to 1.5×IQR. (**C**) We define fluorescence change as the median difference between the fluorescence scores of sequences with vs without a −10 or −35 box. The fluorescence change is only considered valid if there is a significant difference between the central tendency of each distribution based on a two-sided Mann-Whitney U (MWU) test. See methods. (**D**) The change in fluorescence (arbitrary units, a.u.) when gaining or losing −10 and −35 boxes in mutual information hotspots of inactive parent sequences (see *Figure 3—figure supplement 1* for calculation overview). Dashed lines indicate an effect size threshold of ±0.5 a.u. Each circle corresponds to a gain or loss of a box in a mutual information hotspot (see *Figure 2* for hotspots). Circles with letters in parentheses refer to the corresponding Figure panels in *Figure 3—figure supplement 2*. The volume of each violin plot corresponds to a kernel density estimate of each distribution. Data available in *Source data 4*. (**E**) Parent P16-RFP. Top: Mutual information $I_i(b, f)$ between nucleotide identity and fluorescence at each position. Solid red line: mean mutual information, shaded region: ±1 standard deviation when the dataset is randomly split into three equally sized subsets (methods). Bottom: position-weight matrix (PWM) predictions for the −10 boxes (magenta rectangles) and −35 boxes (orange rectangles) along the wild-type parent sequence. (**F**) Top: Parent P16-RFP and its PWM predictions from (**E**). We plot the fluorescence scores of all daughters without a −10 box in the region of interest (left, gray rectangle). Bottom: the most frequent genotype in the dataset where a −10 box is in the region of interest. We plot the fluorescence scores of all daughters with a −10 box in the region of interest. We tested the null hypothesis that the gain of the −10 box significantly increases fluorescence (two-tailed Mann-Whitney U [MWU] test). The q-values correspond to Benjamini-Hochberg-corrected p-values (methods) (two-tailed MWU test, q=7.64 × 10⁻³²). Within the violin plot is a box plot, where the box represents the interquartile range (IQR) and the white circle the median. Whiskers correspond to 1.5×IQR. (**F'**) Top: the fluorescence readouts of three colonies harboring the wild-type P16-RFP reporter measured using a plate reader. The horizontal axis shows the RFP readout normalized to the optical density of the culture (OD₆₀₀) during the reading (see methods). Each point corresponds to the fluorescence of an individual colony. Whiskers correspond to the minimum and maximum values and the dark square the mean fluorescence level. Bottom: analogous to top, but for three colonies harboring a single point mutation identified to change

*Figure 3 continued on next page*

*Figure 3 continued*

fluorescence in panel F. (**G**) Analogous to (**E**) but for the parent P1-RFP. (**H**) Analogous to (**F**) except for gaining a –35 box in the region of interest (two-tailed MWU test, q=4.25 × 10$^{-42}$ (**H′**) Analogous to F′ but for the point mutation identified to change fluorescence in panel H). See ***Figure 3—figure supplement 2*** for additional examples.

The online version of this article includes the following figure supplement(s) for figure 3:

**Figure supplement 1.** Identifying where motifs are gained and lost in hotspots that are associated with changes in fluorescence.

**Figure supplement 2.** Additional examples of which gaining –10 and –35 boxes creates de-novo promoters.

does. Mutations also destroy –10 and –35 boxes originally in the parent sequences, but this does not create de novo promoter activity (see ***Figure 3D***).

## Destroying -10 and -35 boxes in promoters exclusively lowers promoter activity

We next asked how gaining and losing –10 and –35 boxes can change expression driven by parent sequences that have promoter activity. To this end, we repeated the previous analysis but for the parents with promoter activity. See ***Source data 4*** for the coordinates, fluorescence changes, and significance for the –10 and –35 boxes.

We first identified 5 and 3 hotspots in which losing a –10 box or –35 box *decreases* expression (***Figure 4A***). These particular hotspots revealed where promoters are located in each parent. For example, for parent P12-RFP (***Figure 4—figure supplement 1C***), losing a –35 box decreases fluorescence by –1.57 a.u. (two-tailed MWU test, q=6.28 × 10$^{-67}$; ***Figure 4—figure supplement 1D***), and losing a –10 box decreases fluorescence by –1.41 a.u. (two-tailed MWU test, q=6.28 × 10$^{-91}$; ***Figure 4—figure supplement 1E***). Scrambling these motif sequences also lowers fluorescence by 62.6-fold and 49.7-fold respectively (***Figure 4—figure supplement 1D′, E′***). Based on these data and the spacing (17 bp) of these two motifs, we can infer that they constitute the promoter in P12-RFP. See ***Figure 4—figure supplement 1*** for additional examples. We did not find any hotspots in which destroying a –10 box or a –35 box increases expression by more than 0.5 a.u. (see ***Figure 4A***).

## Gaining new motifs over existing motifs increases and decreases promoter activity

Our analysis found that across the 10 parents with promoter activity, 329 new –10 boxes and 332 new –35 boxes are created at unique positions within their daughter sequences. Despite this extensive gain of new boxes, only 4 of these new –10 boxes and 1 of these new –35 boxes further increases expression (***Figure 4B***). More surprisingly, for 4 and 6 of the new –10 and –35 boxes, gaining the box *decreases* expression (n=4 + 1+4 + 6, 15 hotspots with increased or decreased expression; ***Figure 4B***). To understand how this is possible, we examined each of these hotspots in detail.

We found that these mutations frequently create new boxes overlapping those we had identified as part of a promoter (***Figure 4—figure supplement 1***). This occurs when mutations create a –10 box overlapping a –10 box, a –35 box overlapping a –35 box, a –10 box overlapping a –35 box, or a –35 box overlapping a –10 box. We call the resulting event a 'homo-gain' when the new box is of the same type as the one it overlaps, and otherwise a 'hetero-gain' (***Figure 4C***). In either case, the creation of the new box does not always destroy the original box.

6 of the 15 hotspots (40%) show homo-motif gains. For example in P12-RFP (***Figure 4D***), a point mutation creates both a new –10 box that is +1 bp further from the –35 box, but also increases the PWM score of the original –10 box to match the consensus (TACAAT→ TAtAAT). This point mutation increases median expression by +0.78 a.u. (***Figure 4E***); (two-tailed MWU test, q=5.85 × 10$^{-13}$). A construct harboring this point mutation also exhibits a 0.3-fold increase in fluorescence compared to the wild-type (***Figure 4E′***). 3 of the 15 hotspots (20%) show hetero-motif gains. For example, for parent P22-GFP, where our mutational data had allowed us to map the active –35 and –10 boxes (***Figure 4F***), mutations can change the –35 box into a –10 box, which decreases expression by –0.90 a.u. (***Figure 4G***; two-tailed MWU test, q=7.29 × 10$^{-7}$). A construct harboring this point mutation also exhibits a 13-fold decrease in fluorescence compared to the wild-type (***Figure 4G′***). Thus, while the loss in expression appears to be caused by gaining a –10 box, this gain also entails the loss of the

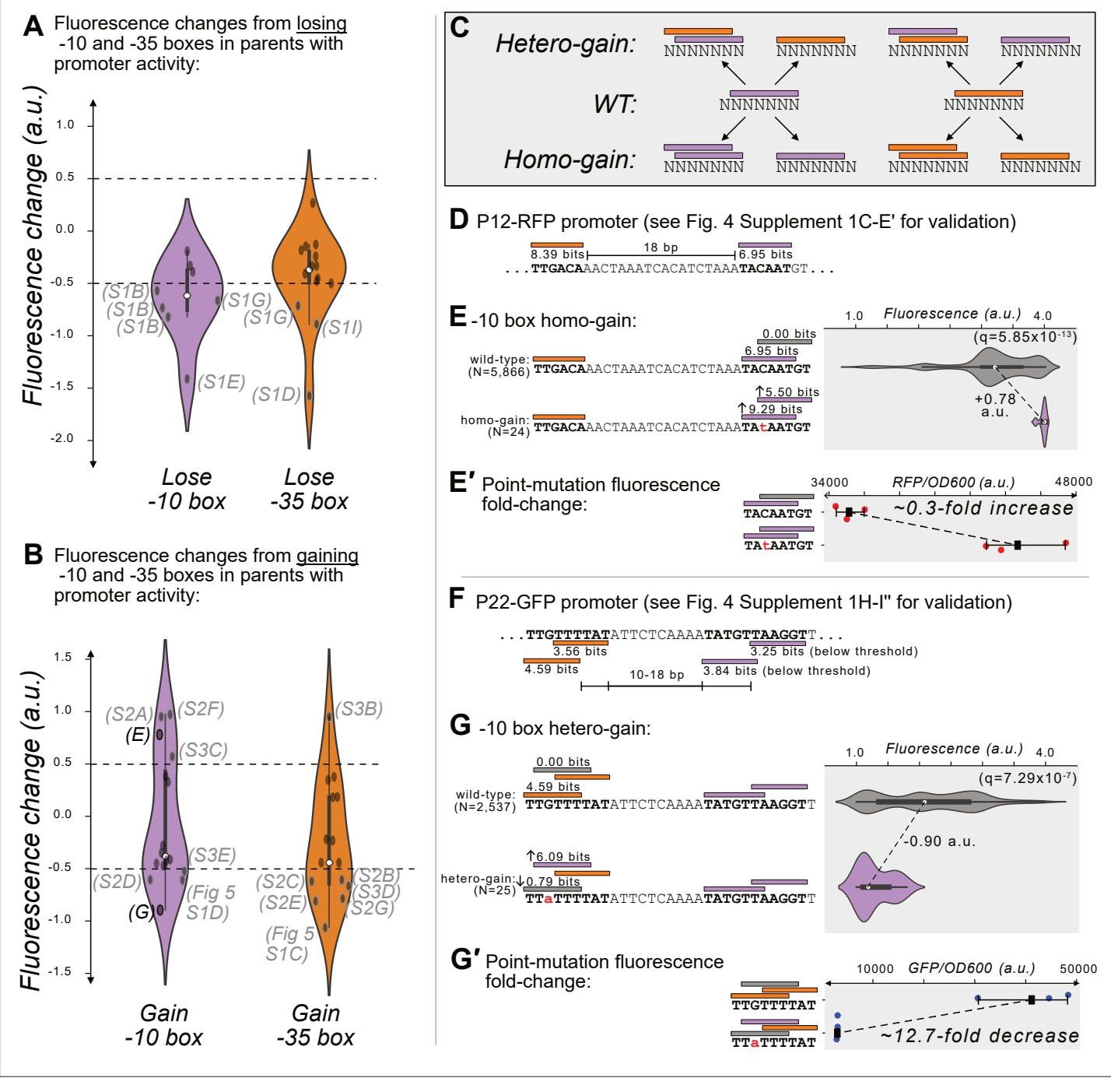

**Figure 4.** Gaining –10 and –35 boxes modulates promoter activity. (**A**) The change in fluorescence (arbitrary units, a.u.) when losing –10 and –35 boxes in hotspots in the active parent sequences. Dashed lines indicate an effect size threshold of ±0.5 a.u. Each black point corresponds to a loss of a –10 or –35 box in a mutual information hotspot. Outlined points with letters in parenthesis highlight the corresponding Figure panels in *Figure 4—figure supplement 1* (S1). The areas of the violin plots are the kernel density estimates (KDE) of each distribution. Within each violin plot is a box plot, where the box represents the interquartile range (IQR) and the white circle the median. Whiskers correspond to 1.5×IQR. Data available in *Source data 4*. (**B**) Analogous to (**A**) but for gaining –10 and –35 boxes instead of losing them. Parenthesis highlight the corresponding Figure panels in *Figure 4—figure supplement 2* (S2), *Figure 4- figure supplement 3* (S3), and *Figure 5—figure supplement 1*. (**C**) In parents with promoter activity, mutations frequently create new –10 (magenta rectangles) and –35 (orange rectangles) boxes over preexisting ones. (**D**) A model of the promoter located in parent P12-RFP. Position-weight matrix (PWM) predictions for the –10 boxes (magenta rectangles) and –35 boxes (orange rectangles) along with the wild-type parent sequence. See *Figure 4—figure supplement 1C–E'* for the experiments validating this promoter. (**E**) Top: Parent P12-RFP and its PWM predictions from (**D**). We plot the fluorescence scores of all daughters without a –10 box in the region of interest (left, gray rectangle). Bottom: the most

*Figure 4 continued on next page*

*Figure 4 continued*

frequent genotype in the dataset where a –10 box is in the region of interest. We plot the fluorescence scores of all daughters with a –10 box in the region of interest. We tested the null hypothesis that the gain of the –10 box significantly increases fluorescence (two-tailed Mann-Whitney U [MWU] test). The q-values correspond to Benjamini-Hochberg-corrected p-values (Methods; two-tailed MWU test, q=$5.85 \times 10^{-13}$). Within the violin plot is a box plot, where the box represents the interquartile range (IQR) and the white circle the median. Whiskers correspond to 1.5×IQR. (**E'**) Top: the fluorescence readouts of three colonies harboring the wild-type P12-RFP reporter, whose activity was measured using a plate reader. The horizontal axis shows red fluorescence normalized to the optical density of the culture ($OD_{600}$) during the reading (see Methods). Each point corresponds to the fluorescence of an individual colony. Whiskers correspond to the minimum and maximum values and the dark square indicates the mean fluorescence level. Bottom: analogous to the top, but for three colonies harboring a point mutation identified to change fluorescence in the preceding panel. (**F**) Analogous to (**D**) but for P22-GFP. Note that the two –10 boxes have PWM scores below the 3.98 bits threshold, but destroying these boxes decreases expression (see *Figure 4—figure supplement 1H–I'''* for the experiments validating this promoter). (**G**) Analogous to (**E**) but for gaining a –10 box in the gray highlighted region of interest on the top strand of P22. (**G'**) Analogous to (E') but for the wild-type and consensus mutant in panel (**G**).

The online version of this article includes the following figure supplement(s) for figure 4:

**Figure supplement 1.** Mapping promoters in active parents.

**Figure supplement 2.** Additional examples of mutations modulating promoter activity.

**Figure supplement 3.** Template P20.

---

promoter's –35 box (see *Figure 4F, Figure 4—figure supplement 1I'*). See *Figure 4—figure supplement 2* for additional examples of the other homo- and hetero-motif gains.

4 of the 15 hotspots (~27%) are located in P20-GFP/RFP (*Figure 4—figure supplement 3*), which encodes a bidirectional promoter. However, because our mutational data did not allow us to confidently map this promoter on either strand of P20, we cannot resolve why gaining –10 or –35 boxes may increase expression (2/4 hotspots) or decrease expression (2/4 hotspots, *Figure 4—figure supplement 3*) of these parents.

To summarize, parents with promoter activity readily gain hundreds of additional –10 and –35 boxes, but this appears to primarily modulate expression only when the new –10 and –35 boxes overlap with the existing promoter.

## Histone-like nucleoid-structuring protein (H-NS) represses P12-RFP and P22-GFP

The histone-like nucleoid structuring protein (H-NS) is a global transcriptional regulator (*Dillon and Dorman, 2010*; *Hommais et al., 2001*; *Martínez-Antonio and Collado-Vides, 2003*). H-NS binds to ~5% of the *E. coli* genome (*Shimada et al., 2011*), to horizontally transferred DNA (*Navarre et al., 2007*; *Oshima et al., 2006*), to transposons (*Cooper et al., 2024*), and to promoter islands (*Purtov et al., 2014*). To test whether H-NS represses the parent sequences, we transformed the parents into a *Δhns* background (KEIO collection, JW1225; *Baba et al., 2006*) and compared their fluorescence levels with those in the DH5α (wild-type) background using a plate reader (see Methods and *Source data 6*).

We plot the fluorescence changes in *Figure 5A* as distributions for the 50 parents, where positive and negative values correspond to an increase or decrease in fluorescence in the *Δhns* background, respectively. Based on the null hypothesis that the parents are not regulated by H-NS, we classified outliers in these distributions (1.5×the interquartile range) as H-NS-target candidates. We refer to these outliers as 'candidates' because the fluorescence changes could also result from indirect *trans*-effects from the knockout (*Mattioli et al., 2020*; *Metzger et al., 2016*). This approach identified six candidates for H-NS targets (P2-GFP, P19-GFP, P20-GFP, P22-GFP, P12-RFP, and P20-RFP). For GFP, the largest change occurs in P22-GFP, increasing fluorescence ~1.6-fold in the mutant background (two-tailed t-test, p=$1.16 \times 10^{-8}$; *Figure 5B*). For RFP, the largest change occurs in P12-RFP, increasing fluorescence ~0.5-fold in the mutant background (two-tailed t-test, p=$4.33 \times 10^{-10}$; *Figure 5B*).

We note that for template P20, which is a bidirectional promoter, GFP expression increases ~2.6-fold in the *Δhns* background (two-tailed t-test, p=$1.59 \times 10^{-6}$). Simultaneously, RFP expression decreases ~0.42-fold in the *Δhns* background (two-tailed t-test, p=$4.77 \times 10^{-4}$; *Figure 5—figure supplement 1A*). These findings suggest that H-NS also modulates the directionality of P20's bidirectional promoter through either cis- or trans-effects.

We hypothesized that H-NS binds to the candidates we had identified as candidate H-NS targets. To validate this hypothesis computationally, we first acquired a PWM for H-NS derived from

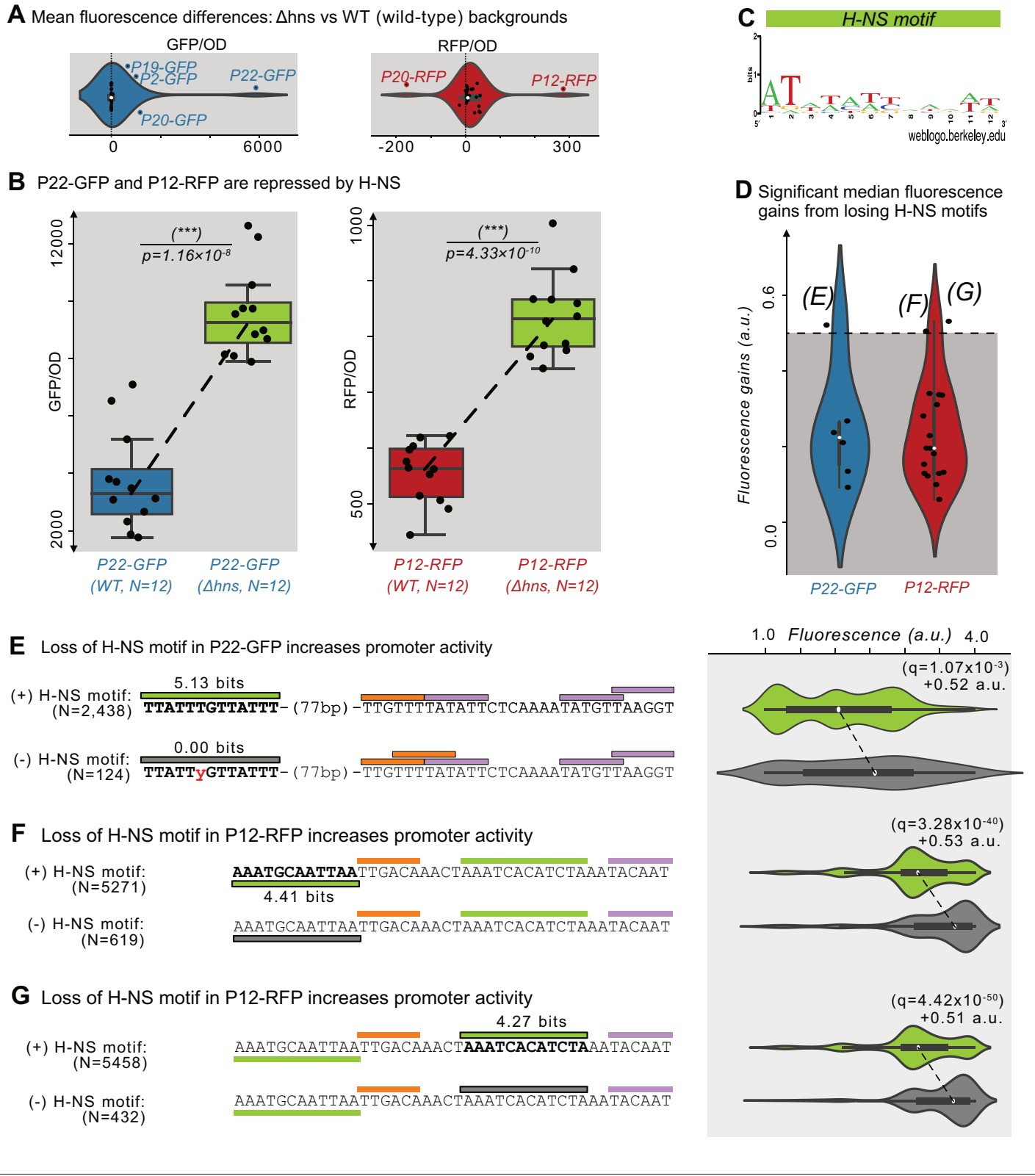

**Figure 5.** Histone-like nucleoid-structuring protein (H–NS) represses P12-RFP and P22-GFP. (**A**) The difference in average fluorescence levels for each parent sequence in a mutant background for the Histone-like nucleoid-structuring protein (H–NS) Δhns vs the wild-type background. Positive and negative values correspond to increases and decreases, respectively in fluorescence in the Δhns background. Fluorescence values measured using a plate reader (see Methods). Within the violin plot is a box plot, where the box represents the interquartile range (IQR) and the white circle the median.

*Figure 5 continued on next page*

*Figure 5 continued*

Whiskers correspond to 1.5×IQR. We classified parents outside of the whiskers as H-NS targets, which are outlined in blue or red, and labeled. Left: GFP fluorescence changes (N=25 parents). Right: RFP fluorescence changes (N=25 parents). (B) The fluorescence levels of bacterial colonies (N=12 each) harboring a parent sequence in the wild-type (left) vs the Δhns background. Box and whisker plots described in (A). We test the null hypothesis that the means in each background are the same using a two-tailed t-test. Left: P22-GFP levels in the wild-type vs the Δhns backgrounds (two-tailed t-test, p=1.16 × 10⁻⁸). Right: P12-RFP levels (two-tailed t-test, p=4.33 × 10⁻¹⁰). (C) A sequence logo derived from a position weight matrix for the transcription factor H-NS (*Tierrafría et al., 2022*). (D) The increase in fluorescence (arbitrary units, a.u.) when losing a H-NS motif in mutual information hotspots of P22-GFP and P12-RFP (see *Figure 3—figure supplement 1* for calculation overview and *Source data 7*). Dashed lines indicate an effect size threshold of +0.5 a.u. Each circle corresponds to a loss of a H-NS motif in a mutual information hotspot (see *Figure 2* for hotspots). Outlined points with letters in parenthesis highlight the corresponding Figure panels. See *Figure 5—figure supplement 1B* for additional parents. (E) Top: Parent P22-GFP and its PWM predictions (orange = –35 box, magenta = –10 box, green = H NS motif of interest). We plot the fluorescence scores of all daughters with the H-NS motif. Bottom: the most frequent genotype in the dataset where the H-NS motif is absent. We plot the fluorescence scores of all daughters without the H-NS motif of interest. We tested the null hypothesis that losing the motif significantly increases fluorescence (two-tailed Mann-Whitney U [MWU] test). The q-values are Benjamini-Hochberg-corrected p-values (methods) (two-tailed MWU test, q=1.07 × 10⁻³). Within the violin plot is a box plot, where the box represents the interquartile range (IQR) and the white circle the median. Whiskers correspond to 1.5×IQR. (F,G) Analogous to (E) but for the highlighted H-NS motif in P12-RFP.

The online version of this article includes the following figure supplement(s) for figure 5:

**Figure supplement 1.** Additional H-NS sites.

**Figure supplement 2.** The change in fluorescence when gaining and losing UP-like motifs.

RegulonDB (*Tierrafría et al., 2022*; *Figure 5C*). We then searched for H-NS-specific hotspots within the 6 parent sequences, that is locations in which mutations that destroy H-NS motifs lead to significant fluorescence increases (Mann-Whitney U-test, *Figure 3—figure supplement 1* and Methods). This approach identified two candidate motifs in P12-RFP and one in P22-GFP whose destruction are associated with fluorescence increases greater than 0.5 a.u. (*Figure 5D*). For the four remaining H-NS target candidates, no H-NS motifs are associated with fluorescence increases of more than 0.5 a.u. (*Figure 5—figure supplement 1B*). See *Source data 7* for the coordinates, fluorescence changes, and significance for the identified H-NS motifs.

For P22-GFP, a H-NS motif lies 77 bp upstream of the mapped promoter. Mutations which destroy this motif significantly increase fluorescence by +0.52 a.u. (two-tailed MWU test, q=1.07 × 10⁻³; *Figure 5E*). For P12-RFP, one H-NS motif lies upstream of the mapped promoter's –35 box, and the other upstream of the mapped promoter's –10 box. Mutations that destroy these H-NS motifs significantly increase fluorescence by +0.53 and+0.51 a.u., respectively (two-tailed MWU test, q=3.28 × 10⁻⁴⁰ and q=4.42 × 10⁻⁵⁰; *Figure 5F and G*). Based on these findings, we conclude that these motifs are bound by H-NS. These findings ultimately demonstrate that H-NS can, in certain contexts, suppress promoter emergence (*Forrest et al., 2022*).

## The binding of H-NS changes when new -10 and -35 boxes are gained

The H-NS sites in P12-RFP also overlap with 2 hotspots for which gaining a new –35 box (TTTTTT→TTTwwT) and a –10 box (TAAATC→TAAATt) significantly decreases fluorescence by –1.1 and –0.61 a.u. (two-tailed MWU test, q=3.35 × 10⁻⁵, q=4.36 × 10⁻⁴; see *Figure 5F and G*, *Figure 5—figure supplement 1C, D*). Since these new –10 and –35 boxes do not overlap with the promoter we mapped on P12-RFP, the reduced fluorescence suggests that promoter motifs can repress promoter activity. However, we could not rule out an alternative hypothesis, namely that these mutations actually modulate H-NS binding.

To distinguish these two hypotheses, we engineered a construct with the new –35 box, and compared its fluorescence with that of wild-type P12-RFP. We observed a significant decrease in fluorescence (two-tailed t-test, p=5.95 × 10⁻⁷; *Figure 5—figure supplement 1E*), confirming the previously observed association (see *Figure 5—figure supplement 1C*). This construct, however, also has significantly higher fluorescence in the Δhns background compared to the wild-type background (two-tailed t-test, p=0.0036; *Figure 5—figure supplement 1E*). In addition, fluorescence of the mutant construct in the Δhns background does not significantly differ from that of P12-RFP in the wild type background (two-tailed t-test, p=0.276). These data show that the Δhns background rescues the fluorescence loss associated with gaining this new –35 box. Thus, expression lowers because mutations modulate H-NS binding, not by gaining the –35 box.

We repeated these experiments with a construct harboring the new –10 box. We observed a significant decrease in fluorescence compared to the wild-type P12-RFP (two-tailed t-test, p=0.0075; *Figure 5—figure supplement 1F*), confirming the previously observed association (*Figure 5—figure supplement 1D*). This construct, however, also has significantly higher fluorescence in the Δ*hns* vs the wild-type background (two-tailed t-test, p=0.0096; *Figure 5—figure supplement 1F*). In addition, fluorescence of the mutant construct in the Δ*hns* background did not differ from that of P12-RFP in the wild-type background (two-tailed t-test, p=0.310). These data show that the Δ*hns* background also rescues the fluorescence loss associated with gaining a new –10 box. Thus, expression lowers because mutations modulate H-NS binding, not by gaining the –10 box.

To summarize, we present evidence that H-NS represses both P22-GFP and P12-RFP in cis. H-NS also modulates the bidirectionality of P20-GFP/RFP in cis or trans. In P22-GFP, the strongest H-NS motif lies upstream of the promoter. In P12-RFP, the strongest H-NS motifs lie upstream of the –10 and –35 boxes of the promoter. We note that there are 16 additional H-NS motifs surrounding the promoter in P12-RFP that may also regulate P12-RFP (*Figure 5—figure supplement 1G*). Mutations in two of these two H-NS motifs can create additional –10 and –35 boxes that appear to lower expression. However, the effects of these mutations are insignificant in the absence of H-NS, suggesting that these mutations actually modulate H-NS binding.

## Single mutations in UP-elements do not strongly contribute to promoter emergence

The UP element is an additional AT-rich promoter motif that can lie stream of a –35 box in a promoter sequence (*Estrem et al., 1998*; *Ross et al., 1993*). We asked whether the creation of UP-elements also creates or modulates promoter activity in our dataset. To this end, we first identified a previously characterized position-weight matrix for the UP element (NNAAAWWTWTTTTNNWAAASYM, PWM threshold score = 19.2 bits; *Estrem et al., 1998*; *Figure 5—figure supplement 2A*). We then computationally searched for UP-element-specific hotspots within the parent sequences, that is locations in which mutations that gain or lose UP-elements lead to significant fluorescence increases (Mann-Whitney U-test, *Figure 3—figure supplement 1* and Methods. See *Source data 8* for the coordinates, fluorescence changes, and significance). The analysis did not identify any UP elements whose mutation significantly changes fluorescence.

We then repeated the analysis with a less stringent PWM threshold of 4.8 bits (1/4th of the PWM threshold score). This time, we identified 74 'UP-like' elements that are created or destroyed at unique positions within the parents. 23 of these motifs significantly change fluorescence when created or destroyed. However, even with this liberal threshold, none of these UP-like elements increase fluorescence by more than 0.5 a.u. when gained, or decrease fluorescence by more than 0.5 a.u. when lost (*Figure 5—figure supplement 2B*). This finding ultimately suggests that the UP element plays a negligible role in promoter emergence within our dataset.

## Discussion

We found that non-promoter parents enriched with –10 and –35 boxes vary dramatically (more than 200-fold) in the probability $P_{new}$ that their mutants will have promoter activity ($P_{new}$ = 0.002–0.41, see *Figure 1D*). When trying to understand the reasons for this variation, we found that $P_{new}$ is not significantly correlated with the number of existing –10 or –35 boxes (least squares linear regression, p=0.515, 0.652, respectively). Instead, we present evidence that promoter emergence is best predicted by the level of background transcription each non-promoter parent produces, a phenomenon also referred to as 'pervasive transcription' (*Kapranov et al., 2007*). From an evolutionary perspective, this would suggest that sequences that produce such pervasive transcripts – including the promoter islands (*Panyukov and Ozoline, 2013*) and the antisense strand of existing promoters (*Dornenburg et al., 2010*; *Warman et al., 2021*), may have a proclivity for evolving de-novo promoters compared to other sequences (*Kapranov et al., 2007*; *Wade and Grainger, 2014*).

A previous study randomly mutagenized the *appY* promoter island upstream of a GFP reporter, and isolated variants with increased and decreased GFP expression. The authors found that variants with higher GFP expression acquired mutations that (1) improve a –10 box to better match its consensus, and simultaneously (2) destroy other –10 and –35 boxes (*Bykov et al., 2020*). The authors

concluded that additional −10 and −35 boxes repress expression driven by promoter islands. Our data challenge this conclusion in several ways.

First, we find that only ~13% of −10 and −35 boxes in promoter islands actually contribute to promoter activity. Extrapolating this percentage to the *appY* promoter island, ~87% (100%−13%) of the motifs would not be contributing to its activity. Assuming the *appY* promoter island is not an outlier, this would insinuate that during random mutagenesis, these inert motifs might have accumulated mutations that do not change fluorescence. Indeed, *Bykov et al., 2020* also found that a similar frequency of −10 and −35 boxes were destroyed in variants selected for lower GFP expression, which supports this argument. Second, we find no evidence that creating a −10 or −35 box lowers promoter activity in any of our 50 parent sequences. Third, we also find no evidence that destruction of a −10 or −35 box increases promoter activity without plausible alternative explanations, that is overlap of the destroyed box with a H-NS site, destruction of the promoter, or simultaneous creation of another motif as a result of the destruction. In sum, −10 and −35 boxes are not likely to repress promoter activity.

For the parents with promoter activity, we found that mutations that create new −10 and −35 boxes only affect transcription if these new boxes partially overlap with the −10 and −35 boxes of the parents' promoters. These new motifs affect expression in various ways. For example, they can change the composition of the spacer between the boxes (*Klein et al., 2021*; *Lagator et al., 2022*; *Urtecho et al., 2019*), create new boxes that better match the consensus sequence of either box (see *Figure 4D*), or even destroy the promoters themselves (see *Figure 4G*). In this way, they can bring forth a range of expression changes. These observations demonstrate how mutations can readily create promoters with a wide dynamic range of activity. This ability facilitates natural selection's fine-tuning of a promoter to precise expression levels, within a small sequence search space that is easily accessible by few DNA mutations. The finding also supports a previous conclusion that true promoters in sequences enriched with promoter motifs typically overlap with multiple −10 and −35 boxes (*Huerta and Collado-Vides, 2003*).

For the 40 parents without promoter activity, we found that mutations readily created over 3,000 new −10 or −35 boxes at unique positions, with 114 of these including 'extended' −10 boxes, and 684 of these comprising a 'standard' promoter with a canonical −35 and −10 box spaced 15–20 bp apart. However, fewer than 9 of these new boxes actually created detectable promoter activity (~0.3% of all new boxes). Additionally, we identified 74 UP-like-elements that were created or destroyed in the dataset, but none of these led to a large significant change in promoter activity.

These findings suggest that we are still underestimating the complexity of promoters. For instance, the −10 and −35 boxes, extended −10, and the UP-element may be one of many components underlying promoter architecture. Other components may include flanking sequences (*Mitchell et al., 2003*), which have been observed to play an important role in eukaryotic transcriptional regulation (*Afek et al., 2014*; *Chiu et al., 2022*; *Farley et al., 2015*; *Gordân et al., 2013*). Recent studies on *E. coli* promoters even characterize an AT-rich motif within the spacer sequence (*Warman et al., 2020*), and other studies use longer −10 and −35 box consensus sequences (*Lagator et al., 2022*). Another possibility is that there is much more transcriptional repression in the genome than anticipated (*Singh et al., 2014*). This would also coincide with the observed repression of H-NS in P22-GFP and P12-RFP, and accounts of H-NS-repression in the full promoter island sequences (*Purtov et al., 2014*).

Our observations call for future large-scale experiments like ours, together with more complex computational models of promoters, such as thermodynamic models, and computational tools such as machine learning to help us understand promoter architecture. (*LaFleur et al., 2022*; *Lagator et al., 2022*; *Urtecho et al., 2023*; *Wang et al., 2020*). Additionally, future studies will be necessary to address the limitations of our own work. First, we use binary thresholding to determine (i) the presence or absence of a motif, (ii) whether a sequence has promoter activity or not, and (iii) whether a part of a sequence is a hotspot or not. While chosen systematically, the thresholds we use for these decisions may cause us to miss subtle but important aspects of promoter evolution and emergence. Second, we measure protein expression through fluorescence as a readout for promoter activity. This readout combines transcription and translation. This means that we cannot differentiate between transcriptional and post-transcriptional regulation, including phenomena such as premature RNA termination (*Song et al., 2022*; *Uptain and Chamberlin, 1997*), post-transcriptional modifications (*Mohanty and Kushner, 2006*), and RNA-folding from riboswitch-like sequences (*Mandal and Breaker, 2004*).

Overall, our study demonstrates that −10 and −35 boxes neither prevent existing promoters from driving expression, nor do they prevent new promoters from emerging by mutation. It shows how mutations can create new −10 and −35 boxes near or on top of preexisting ones to modulate expression. However, randomly creating a new −10 or −35 box will rarely create a new promoter, even if the new box is appropriately spaced upstream or downstream of a cognate box. Ultimately our study demonstrates that promoter models need to be further scrutinized, and that using mutagenesis to create de-novo promoters can provide new insights into promoter regulatory logic.

## Methods

### Position weight matrices (PWMs)

PWMs for the −10 and −35 boxes are from *Belliveau et al., 2018*, who derived the instances from 145 of the top predicted −10 and −35 boxes from the database, RegulonDB (*Tierrafría et al., 2022*). We converted them into PWMs using Biopython.motifs from the Biopython package (*Cock et al., 2009*) (v1.79). We deliberately assumed a uniform nucleotide composition (25% A, T, C, or G) in this process, because doing so allowed us to test PWM prediction performance in a variety of different sequence backgrounds. For every input sequence, a PWM returns an output score in bits. The higher this score is, the more likely it is that the input sequence can be bound by the protein the PWM represents.

To classify whether an output score is strong enough to be classified as a −10 or −35 box, we used the standard practice of setting a threshold equal to the information content of the PWM (*Hertz and Stormo, 1999*). The −10 box in our study has an information content of 3.98 bits and the −35 box of 3.39 bits. We classified input sequences with scores at least this high as −10 or −35 box, respectively. In *Figure 4—figure supplement 1* we also plot 'low-affinity' −10 and −35 boxes. For these motifs, we halved the thresholds (1.99 and 1.70 bits, respectively).

### Plasmid MR1 (pMR1)

The plasmid MR1 (pMR1) is a variant of the plasmid RV2 (pRV2) in which the *kan* resistance gene has been swapped with the *cm* resistance gene (*Guazzaroni and Silva-Rocha, 2014*). Plasmid pMR1 encodes the BBa_J34801 ribosomal binding site (RBS, AAAGAGGAGAAA) 6 bp upstream of the start codon for GFP(LVA). The plasmid also encodes a putative RBS (AAGGGAGG; *Cazemier et al., 1999*) 5 bp upstream of the start codon for mCherry on the opposite strand. The plasmid additionally contains the low-to-medium copy number origin of replication p15A (*Westmann et al., 2018*). A map of the plasmid is available on the Github repository: https://github.com/tfuqua95/promoter_islands (copy archived *Fuqua, 2024*).

### Amplifying parent sequences and plasmid MR1 with PCR

We acquired the genomic coordinates for the promoter island sequences from *Panyukov and Ozoline, 2013*. We used a polymerase chain (PCR) reaction to amplify 78 parent sequences from the genome, using Q5 polymerase (NEB, USA product #M0491). The reaction amplifies each parent sequence from the genome, and additionally concatenates short sequences to the 5′ and 3′ ends of the inserts (5′-GGCTGAATTC…insert…GGATCCTTGC-3′). These overhangs allowed us to carry out the error-prone PCR with a single set of primers. See *Source data 1* for the list of primers and *Figure 1—figure supplement 6* for an overview of the PCR.

To amplify the parent sequences from the genome, we performed each PCR in a 50 µL reaction volume with the following reagents: 1 µL of each primer (2×primers, 2 µL total, concentration 100 µMol), 5 µL Q5 reaction buffer, 1 µL 10 mM dNTPs (Thermo Fisher Scientific, USA, product #R0191), 1 µL of the genomic DNA extracted using a Wizard Genomic DNA Purification Kit according to the manufacturer's instructions (A1120, Promega USA), 1 µL Q5 high-fidelity DNA polymerase, and 40 µL of water. We carried out the PCR in a thermal cycler (C1000 Touch Thermal Cycler, Bio-Rad, USA) with 30 cycles at 55 °C for annealing and 72 °C for primer extension, both for 30 s. To purify the PCR products, we separated them on a 1% agarose gel, excised them, and used a Qiagen QIAquick Gel Purification Kit (QIAGEN, Germany, product #28706) following the manufacturer's instructions. To verify the purification's success and estimate the final concentrations of each product, we used a Nanodrop One spectrophotometer (Thermo Fisher Scientific, USA).

To create the mutagenesis library of parent sequences, we pooled the 78 PCR products such that each product had the same concentration in the pool. We then performed a single error-prone polymerase chain reaction on the pooled products to create a mutagenesis library. To this end, we used GoTaq polymerase (Promega, USA, product #M3001). Specifically, we combined 1 μL of two PCR primers complementary to both the pMR1 plasmid and to the sequences concatenated to the ends of the initial PCR (1 μL forward and 1 μL reverse primer, each at a 100 μMol l$^{-1}$; see *Source data 1* and *Figure 1—figure supplement 6*), 10 μL of GoTaq reaction buffer, 1 μL 10 mM dNTPs (Thermo Fisher Scientific, USA, product #R0191), 1 μL of pooled DNA, and 1 μL of GoTaq polymerase. We added water to a total of 98 μL, then added 2 μL of 15 mMol MnCl$_2$ to the reaction. We then excised and purified the product as described for the parent sequences.

We additionally used Q5 polymerase to create and amplify linear copies of pMR1 (see *Source data 1* for primer sequences and *Figure 1—figure supplement 6*). We carried out the necessary PCR as described for the parent sequences but for an extension time of 2 min and 30 s.

## Molecular cloning

We cloned the error-prone PCR library into pMR1 at the EcoRI and BamHI sites using NEBuilder (New England Biolabs, USA, product #E2621) as follows: 100 ng of linear pMR1, 25 ng insert, 5 μL NEBuilder mastermix, and water to 10 μL total volume. To increase the reaction efficiency, we added 1 μL of T4 DNA ligase buffer (New England Biolabs, USA, product #B0202S). Using a thermal cycler (C1000 Touch Thermal Cycler, Bio-Rad, USA) we incubated each assembly reaction for 1 hr at 50 °C.

Subsequently, we electroporated our product directly into *E. coli* DH5α electrocompetent cells (Takara, Japan, product #9027) by adding 2 μL of (unpurified) product to 2×50 μL aliquots of *E. coli* (100 μL total) in a 2 mm electroporation cuvette (Cell Projects, England, product #EP-202), and electroporated the cells using a Bio-Rad MicroPulser (Bio-Rad, USA). For recovery, we then added 1 mL of Super Optimal Broth with Catabolite Repression Medium (SOC, provided with the electrocompetent cells (Takara, Japan, product #9027)), shaking the cells at 230 RPM at 37 °C for 1.5 hr (Infors HT, Switzerland, Multitron).

We then plated 5 μL of the bacterial culture onto a petri dish containing LB-agar supplemented with 100 μg/ml of chloramphenicol, and incubated the plate at 37 °C overnight. In parallel, we added 9 mL of LB-chloramphenicol (100 μg/mL) to the remaining 995 μL of bacteria (in SOC), and incubated the resulting culture overnight. The following morning, we manually counted colonies on one quarter of the plate, and estimated the total number of clones (cells) in the original electroporated culture by multiplying this value by 4×200.

## PCR-stitching for validation constructs

To validate the predictions of our random mutagenesis experiments, we created variants of inserts in our expression reporter plasmids pMR1 harboring either (1) individual point mutations in a promoter island, or (2) scrambled motif sequences within the promoter island fragments. To design the motifs for scrambling, we used the random.shuffle function from the python random library to maintain both the GC-content of the region and the spacing of surrounding elements. After computationally shuffling each motif, we checked that the shuffling did not result in the creation of a new –10 or –35 motif. If it did, we repeated the shuffling until it did not. We then proceeded to create the mutants in-vivo.

For each insert, we carried out two PCRs. For the first PCR, we use the forward primer 'pMR1_construct_gibson_assembly_forward', and a specific reverse primer labeled 'bottom' which contains the mutation(s) of interest (primers in *Source data 1*). The bottom primer includes either the point mutations or the scrambled motif sequences. For the second PCR, we used the reverse primer 'pMR1_construct_gibson_assembly_reverse', and a specific forward primer labeled 'top' in *Source data 1*, which is the reverse complement of the bottom primer (primers in *Source data 1*). For both PCRs, the amplification template was 1 μL from a mini prep of the template sequence (P1-P25) already cloned into pMR1 as described in 'Amplifying parent sequences and plasmid MR1 with PCR'.

Because we used plasmids harboring the template sequence for PCR amplification, the product of each PCR includes inserts with the mutations and others with the wild-type inserts. To remove the template from the reaction mixture, we directly added 5 μL of rCutsmart buffer (NEB, USA, product #B6004S) and 1 μL of DpnI (NEB, USA, product #R0176S) to each reaction mix, and incubated the reactions at 36 °C for 1 hr followed by a 70 °C incubation for 15 min using a thermocycler (C1000

Touch Thermal Cycler, Bio-Rad, USA). We then purified the products using a Qiagen QIAquick Gel Purification Kit (QIAGEN, Germany, product #28706), as described by the manufacturer.

We assembled the first and second PCR products with the pMR1 plasmid using 100 ng of linear pMR1, 25 ng of the first PCR product, 25 ng of the second PCR product, 5 µL NEBuilder mastermix, and water to a total volume of 10 µL (New England Biolabs, USA, product #E2621). The assembly and transformations were then carried out as described in 'Molecular cloning'. To confirm the mutations, we selected a single colony and used the primer pMR1_F2 to sequence the insert by Sanger sequencing.

## Measuring fluorescence with a plate reader

We streaked bacteria from glycerol stocks onto LB-agar plates with the appropriate selection antibiotics. For the plasmids that we transformed into the DH5α strain (wild-type), this included 100 µg/ml of chloramphenicol. For the plasmids that we transformed into the Δhns strain, this included the 100 µg/ml of chloramphenicol with an additional 50 µg/ml of kanamycin. We incubated the plates overnight (~16 hr) at 37 °C. We then selected individual colonies and inoculated them into 200 µL cultures of LB with 100 µg/ml of chloramphenicol at the aforementioned concentrations. We grew the cultures overnight (~16 hr) at 37 °C, shaking at 230 RPM with an Infors HT incubator (Multitron, Switzerland).

We centrifuged the cultures for 10 min at 3200 g (Eppendorf, Germany, #5810 R), removed the supernatants, and resuspended the pellets in 200 µL of Dulbeco's Phosphate Buffer Solution (PBS; Sigma, USA, D8537). We repeated this wash step, and resuspended the pellets once again in 200 µL of Dulbeco's PBS.

We pipetted 200 µL of the cultures into a 96-well microplate (Sarstedt AG, Germany, #3924) and measured the fluorescence using a Tecan Spark plate reader (Tecan, Switzerland). We measured GFP with an excitation of 485 nm (bandwidth 20 nm) and emission of 535 nm (bandwidth 20 nm). We measured red fluorescence with an excitation of 560 nm (bandwidth 20 nm) and emission of 620 nm (bandwidth 20 nm). We additionally measured the optical density at 600 nm (bandwidth 3.5 nm; $OD_{600}$). We report all fluorescence readings from the plate reader using the fluorescence counts normalized to the $OD_{600}$. See *Source data 6* for the readings.

## Sort-seq controls

We synthesized three control promoter sequences (*Source data 1*) as fiducial markers for defining the fluorescence sorting gates for sort-seq. The insert DNA sequences were chemically synthesized (Integrated DNA Technologies, USA). We then amplified these promoters, cloned them into plasmid MR1, and transformed *E. coli* with them as described in 'Molecular Cloning'. The first is the promoter iGem bba_j23110 on the top DNA strand, which drives moderate GFP expression from plasmid pMR1. The second is the reverse complement of this promoter, which drives moderate RFP expression from the bottom DNA strand of pMR1. The third and final (negative) control is the wild-type pMR1 plasmid without any insert in its multiple cloning sequence.

## Sort-Seq

To perform sort-seq, we first added 100 µL of the glycerol stock of our mutagenesis library to 1 mL of LB-chloramphenicol (100 µg/ml). We then allowed the resulting culture to shake at 230 RPM and 37 °C overnight (~16 hr) on an Infors HT (Multitron, Switzerland) shaking incubator. Subsequently, we centrifuged the culture and washed the cell pellet twice with Dulbeco's Phosphate Buffered Saline (PBS) (Sigma, USA, D8537). We then resuspended the pellet in PBS.

To detect fluorescence caused by GFP and RFP expression, we measured green fluorescence using a 488 nm laser (FITC-H) at 750 volts, and red fluorescence with a 633 nm laser (PE-H) at 510 volts on an Aria III cell sorter (BD Biosciences, USA). We used the following fluorescence gates for sort-seq (*Figure 1—figure supplement 2*).

RFP bin #1 (none, i.e., no fluorescence): We defined a minimum boundary to not sort cell debris, salts, empty droplets, and other impurities, as the larger of two PE-H values. These values are the minimum PE-H of the negative control (empty pMR1 plasmid) and the minimum PE-H of the opposite fluorophore positive control (GFP in this case). The upper boundary of this bin is the highest PE-H value detected for these same controls.

RFP bin #4 (high): We defined the lower boundary of this bin using the mean fluorescence level of the positive RFP control. This bin does not have an upper bound, because it encompasses the highest levels of fluorescence.

RFP bins #2 and #3 (low and moderate): the lower bound of bin #2 begins at the upper bound of bin #1. The upper bound of bin #3 ends at the lower bound of bin #4. The upper bound of bin #2, which is identical to the lower bound of bin #3, equals the average of the lower boundary of bin #4 and the upper bound of bin #1.

We defined the bins for GFP analogously, but with GFP and RFP controls swapped, with the following exception: Because the GFP positive control produces a bimodal FITC-H distribution, we defined the lower bound of bin #4 for green fluorescence as the peak of the higher mode of this distribution. See *Figure 1—figure supplement 2* for an illustration of the bins.

We sorted the library using two consecutive rounds of sorting on consecutive days. On the first day, we sorted 2,160,000, 71,545, 28,399, and 22,720 cells into GFP bins none, low, moderate, and high, respectively; and 2,160,000, 540,000, 101,636, and 76,898 cells into RFP bins none, low, moderate, and high, respectively. After sorting, we added 1 mL of SOC medium (Sigma, USA, product #CMR002K) to the sorted cultures, incubating the cells at 37 °C and shaking for 2 hr at 230 RPM (Infors HT, Switzerland, Multitron). Following this recovery period, we added 9 mL of LB-Chloramphenicol (100 μg/ml chloramphenicol) to each culture and incubated them overnight (~16 hr) under the same conditions.

On day 2, we once again washed the cultures and re-sorted cells in each culture into the same fluorescence bin. That is, we sorted cells from GFP-none into GFP-none, GFP-low into GFP-low, and so forth. This resorting step ensures that each genotype fluoresces within the same boundaries as on the previous day, and reduces the likelihood of propagating sorting errors into subsequent analyses. The resulting number of cells for each bin were as follows: GFP-none: 6,480,000, GFP-low: 216,000, GFP-moderate: 90,000, GFP-high: 69,000, RFP-none: 6,480,000, RFP-low: 540,000, RFP-moderate: 306,000, RFP-high 231,000 cells. We allowed the cells to recover as described at the end of day 1.

After ~16 hr of overnight culture, we created glycerol stocks for each culture (8 cultures total) as described in 'Molecular Cloning'. From the remaining culture volume, we isolated the plasmids using a Qiagen QIAprep Spin Miniprep kit (QIAGEN, Germany, product #27104) as described by the manufacturer. We amplified the plasmid inserts using PCR as described in 'Molecular Cloning', using primers (*Source data 1*) that add a unique multiplexing barcode onto the 5'-end of each PCR product from each bin. In addition, we isolated the inserts from the unsorted library, and endowed them too with a unique barcode. This procedure yielded 9 (=8 + 1) sequence samples, each with a unique barcode. We then pooled equimolar amounts of these samples, and sequenced them using Illumina paired-end sequencing (Eurofins GmbH, Germany).

## Illumina sequencing

The amplicon pool was sequenced by Eurofins Genomics (Eurofins GmbH, Germany) using a NovaSeq 6000 (Illumina, USA) sequencer, with an S4 flow cell, and a PE150 (Paired-end 150 bp) run. In total, 282,843,000 reads and 84,852,900,000 bases were sequenced. Raw sequencing reads can be found here: https://www.ncbi.nlm.nih.gov/bioproject/1071572.

## Processing sequencing reads

Using Flash2 (*Magoč and Salzberg, 2011*), we merged the paired-end reads that resulted from Illumina sequencing. Each paired-end read contains palindromic cut-sites for 5'-EcoRI (GAATTC) and 3'-BamHI (GGATCC). To orient all of the paired-end reads in the same direction, we searched each paired-end read for both cut-sites, discarding any sequence that did not encode both, and took the reverse complement of the read if the BamHI site was upstream of the EcoRI site. We then identified the tagmented barcode sequence upstream of the EcoRI site, classifying from which fluorescence bin the read originated. We then cropped the sequences to remove the barcodes, BamHI, and EcoRI sites, and counted the total number of reads within every bin. We refer to each of these unique paired-end read sequences as *daughter sequences*.

To focus on point mutations rather than insertions and deletions, we removed daughter sequences with a length different from the wild-type 150 bp. To minimize the influence of sequencing errors in

our dataset, we removed daughter sequences that did not occur at least once in the unsorted library, and daughter sequences with less than thirty reads in the entire dataset.

To map each daughter sequence to its respective template parent sequence, we calculated the Hamming distance (i.e. the number of nucleotide differences between two sequences) between each daughter sequence and the 78 parent sequences, and identified the parent as the sequence with the smallest Hamming distance. To minimize our likelihood of mapping the daughter sequences to the wrong parent sequence, we removed daughter sequences from the dataset for which the closest Hamming distance was greater than 10. Because the error-prone PCR created different numbers of daughter sequences per parent sequence, we also removed all parent and their respective daughter sequences if there were less than 2000 unique daughter sequences. After these calculations and filtration steps, our dataset contained 245,639 daughter sequences derived from 25 parent sequences.

To calculate fluorescence scores (*F*) we used in the next analysis step, we calculated (*F*) using **Equation (1)**:

$$F = \frac{\sum_{f=1}^{4} \left( f \times Reads_f \right)}{\sum_{f=1}^{4} \left( Reads_f \right)}$$

(1)

Here, *f* is an integer index corresponding to each of the four fluorescence bins, that is *f*=1,2,3,4 corresponds to no fluorescence, low, moderate, and high fluorescence, respectively. $Reads_f$ is the number of times the sequence was identified in each fluorescence bin *f*. We calculated *F* for both green and red fluorescence. Each score is in arbitrary units (a.u.) of fluorescence.

See **Source data 2** for a csv file with the daughter sequences and their respective fluorescence values, which we analyzed further with the code on the GitHub repository: https://github.com/tfuqua95/promoter_islands, (copy archived **Fuqua, 2024**).

## Mutual information

To calculate mutual information between nucleotides and fluorescence scores, we first randomly split the dataset into three equal-sized pseudo-replicates (r1, r2, and r3) to minimize the effects of sampling biases as well as sorting errors when calculating mutual information. For each pseudo-replicate, we calculated the mutual information $I_i\left(b,f\right)$ between the nucleotide identity *b* and fluorescence score *f* score at each position *i* (1<*i* < 150), of every daughter sequence using **Equation 2**, as defined in **Kinney et al., 2010**:

$$I_i\left(b,f\right) = \sum_b \sum_f p_i\left(b,f\right) \log_2 \frac{p_i\left(b,f\right)}{p_i\left(b\right) \times p\left(f\right)} - \frac{\left(n_b - 1\right)\left(n_f - 1\right) \log_2 e}{2N} + O\left(N^{-2}\right)$$

(2)

In the left term on the right-hand side of **Equation 2**, *b* indexes each nucleotide base (*b*=A,T,C,G) and *f* denotes the fluorescence score (see Processing sequencing reads) rounded to the nearest whole number (*f*=1,2,3,4). The expression $p_i\left(b\right)$ denotes the relative frequency of base *b* (A, T, C, or G) at position *i*. *p(f)* is the relative frequency of a daughter sequence with a (rounded) fluorescence score of 1,2,3, or 4. $p_i\left(b,f\right)$ is the observed joint probability, that is the relative frequency of a daughter sequence having a score of *f* and base *b* at position *i*.

The right term on the right-hand side of **Equation 2** is a correction term for differences in degrees of freedom (9 degrees of freedom), where $n_b$ = 4 is the number of possible bases $n_f$ = 4 is the number of possible (rounded) fluorescence scores, and *N* is the size of the library. The fewer degrees of freedom there are and the larger the library is, the smaller the correction term.

For the mutual information plots shown in the text, we computed a Gaussian filter with the ndimage (alpha = 1) function in scipy (v1.8.1). In these plots, the solid line is always the average mutual information between r1, r2, and r3 at each position, and the lightly shaded region indicates ±1 standard deviation between r1, r2, and r3 at each position.

## Identifying mutual information hotspots

We calculated the mutual information for each parent and its daughters (see Mutual Information), and used the following procedure to automatically identify mutual information hotspots. First, we subtracted the standard deviation from the mean mutual information at each nucleotide position

*i*, and truncated all values smaller than 0.0005 bits to 0 bits. We refer to the resulting quantity as *adjusted* mutual information. We then established the 90th percentile of this adjusted mutual information at each nucleotide position *i* as a threshold to identify mutual information hotspots. Specifically, we identified mutual information hotspots as nucleotide positions where the adjusted mutual information (1) lies above this threshold, and (2) is greater than the mutual information at the positions *i*-1 and *i*+1. See *Source data 3* for hotspot coordinates.

### Analysis of motif gain and loss

To identify hotspots in which gaining and losing a −10 or −35 box associates with a significant change in gene expression (fluorescence), we performed the following procedure. First, we drew a 6 bp sliding window starting at the beginning of each parent sequence. Within this window, we calculated the respective −10 and −35 PWM score for every daughter sequence (see *Figure 3—figure supplement 1A*). For each daughter sequence, if the binding score was greater than a designated threshold (−35 box = 3.39 bits, −10 box = 3.98 bits, see 'Position Weight Matrices'), we added the fluorescence score to a 'Motif (+)' list. If the PWM score was below this threshold, we added the score to a 'Motif (-)' list (see *Figure 3—figure supplement 1B*).

If each list contained more than 10 values, we then tested the null hypothesis that the fluorescence scores have the same central tendency, using a two-sided Mann-Whitney U test (mannwhitneyu function from scipy.stats, v1.8.1). If the p-value was less than 0.05 and was located within ±3 bp from a hotspot, we recorded the value in *Source data 4*. We then advanced the 6 bp window in steps of one nucleotide position until it had reached the final end of the parent sequence, and repeated this calculation at each location of the sliding window. See *Figure 3—figure supplement 1*.

We performed this analysis for both GFP and RFP fluorescence scores on their respective DNA strands for both −10 and −35 boxes and for all 25 parent sequences (*Source data 4*). To account for multiple hypothesis testing, we calculated corrected q-values using a Benjamini-Hochberg correction (false-discovery rate 0.05).

For subsequent analyses, we only focused on significant associations found within ±3 bp from the peak of each hotspot (see 'Identifying promoter emergence hotspots' and 'Mutual information'), and associations where the median fluorescence difference between the lists box present and box absent is larger than the absolute value of 0.5 arbitrary units (a.u.).

We carried out the same analyses for H-NS (*Source data 7*) and the UP-element (*Source data 8*).

### Sigmoid curve-fitting

To fit the distribution of Pnew to a sigmoid function, we used the function curve_fit from the python module scipy.optimize (version 1.8.1). The sigmoid function we employ is based on the equation

$$Pnew = \frac{L}{\left(1 + e^{-k(x_0 - x)}\right)}$$

(3)

Here, L corresponds to the upper asymptote, $x_0$ is the inflection point, k is the slope of the curve, and x corresponds to a fluorescence value measured from a parent sequence. After fitting L, $x_0$, and k, we tested the null hypotheses that the parameters L, $x_0$, and k are equal to 0 using a non-linear least squares regression ($P<Z$, $P<Z$, and $1.43 \times 10^{-9}$ for L, $x_0$, and k), respectively.

## Acknowledgements

The European Molecular Biology Organization (EMBO) supports TF with a postdoctoral fellowship (ALTF 963–2021). TF is also supported by a University of Zurich Postdoc Fellowship (FK-23–120). AW and his group is supported by the Swiss National Science Foundation (grants 310030_208174).

We would like to thank the Wagner lab for all discussions and conversations, even if they were not always pertinent to science. We additionally thank Baxter for inspiring Tim with the analysis pipeline during a painfully slow and cold morning walk.

# Additional information

### Funding

| Funder | Grant reference number | Author |
|---|---|---|
| European Molecular Biology Organization | ALTF 963-2021 | Timothy Fuqua |
| University of Zurich | FK-23-120 | Timothy Fuqua |
| Swiss National Science Foundation | 310030_208174 | Andreas Wagner |

The funders had no role in study design, data collection and interpretation, or the decision to submit the work for publication.

### Author contributions

Timothy Fuqua, Conceptualization, Data curation, Formal analysis, Supervision, Funding acquisition, Validation, Visualization, Methodology, Writing – original draft, Writing – review and editing; Yiqiao Sun, Data curation, Formal analysis, Investigation; Andreas Wagner, Conceptualization, Resources, Formal analysis, Supervision, Funding acquisition, Visualization, Writing – original draft, Project administration, Writing – review and editing

### Author ORCIDs

Timothy Fuqua ⓘ https://orcid.org/0000-0003-4005-3329
Andreas Wagner ⓘ https://orcid.org/0000-0003-4299-3840

Reviewer #1 (Public review): https://doi.org/10.7554/eLife.98654.3.sa1
Reviewer #2 (Public review): https://doi.org/10.7554/eLife.98654.3.sa2
Reviewer #3 (Public review): https://doi.org/10.7554/eLife.98654.3.sa3
Author response https://doi.org/10.7554/eLife.98654.3.sa4

# Additional files

### Supplementary files

• MDAR checklist

• Source data 1. Is a table of the DNA sequences for the primers and template sequences in this study. Data is stored as an Excel spreadsheet.

• Source data 2. Is a dataframe with all daughter sequences and their respective fluorescence scores after quality filtering the data as described in the Methods subsection: Processing sequencing reads. Data is stored as a csv file.

• Source data 3. Is a dataframe containing the location of each mutational information hotspot, and information on whether the hotspot overlaps with a –10 or –35 box. Data is stored as a csv file.

• Source data 4. Is a dataframe with the results from our computational search for hotspots where –10 and –35 boxes are gained or lost, and the extent to which they associate with significant fluorescence changes. Data stored as a csv file.

• Source data 5. Is an Excel spreadsheet that can help to rapidly reproduce the main Figures (to the best of Excel's capabilities).

• Source data 6. Is a dataframe with fluorescence readouts from our plate-reader experiments. Data is stored as a csv file.

• Source data 7. Is a dataframe with the results from our computational search for hotspots where H-NS motifs are gained or lost, and the extent to which they associate with significant fluorescence changes. Data stored as a csv file.

• Source data 8. Is a dataframe with the results from our computational search for hotspots where UP-element motifs are gained or lost, and the extent to which they associate with significant fluorescence changes. Data stored as a csv file.

## Data availability

A Jupyter notebook with all of the python scripts to recreate the analysis and figures in this study, as well as *Source datas 1–8* are available at the Github repository: https://github.com/tfuqua95/promoter_islands (copy archived *Fuqua, 2024*). Sequence reads are available at the Sequence Read Archive (SRA) with accession number: PRJNA1071572 (https://www.ncbi.nlm.nih.gov/bioproject/1071572).

The following dataset was generated:

| Author(s) | Year | Dataset title | Dataset URL | Database and Identifier |
|---|---|---|---|---|
| Wagner S, Fuqua T | 2024 | Promoter Island Sort-Seq | https://www.ncbi.nlm.nih.gov/bioproject/PRJNA1071572 | NCBI BioProject, PRJNA1071572 |

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
