## [Editor Report · eLife Assessment]

This **important** study explores the relationship between the sequence of prokaryotic promoter elements and their activity using mutagenesis to generate thousands of mutant sequences. The evidence supporting these findings is **convincing**. This work will appeal to those interested in bacterial genetics, genome evolution, and gene regulation.

---

## [Referee Report · Reviewer #1 (Public review)]

Summary:

This study by Fuqua et al. studies the emergence of sigma70 promoters in bacterial genomes. While there have been several studies to explore how mutations lead to promoter activity, this is the first to explore this phenomena in a wide variety of backgrounds, which notably contain a diverse assortment of local sigma70 motifs in variable configurations. By exploring how mutations affect promoter activity in such diverse backgrounds, they are able to identify a variety of anecdotal examples of gain/loss of promoter activity and propose several mechanisms for how these mutations are interacting within the local motif landscape. Ultimately, they show how different sequences have different probabilities of gaining/losing promoter activity and may do so through a variety of mechanisms.

Major strengths and weaknesses of the methods and results:

This study uses Sort-Seq to characterize promoter activity, which has been adopted by multiple groups and shown to be robust. Furthermore, they use a slightly altered protocol which allows measurements of bi-directional promoter activity. This combined with their pooling strategy allows them to characterize expression of many different backgrounds in both directions in extremely high-throughput which is impressive! A second key approach this study relies on is the identification of promoter motifs using position weight matrices (PWMs). While these methods are prone to false positives, the authors implement a systematic approach which is standard in the field. However, drawing these types of binary definitions (is this a motif? yes/no) should always come with the caveat that gene expression is quantitative traits that we oversimplify when drawing boundaries.

Their approach to randomly mutagenize promoters allowed them to find many examples of different types of evolutions that may occur to increase or decrease promoter activity. They have supported these with validations in more controlled backgrounds which convincingly support their proposed mechanisms for promoter evolution.

An appraisal of whether the authors achieved their aims, and whether the results support their conclusions:

The authors express a key finding that the specific landscape of promoter motifs in a sequence affect the likelihood that local mutations create or destroy regulatory elements. The authors have described many examples, including several that are non-obvious, and show convincingly that different sequence backgrounds have different probabilities for gaining or losing promoter activity. This overarching conclusion is supported by trend and mechanistic data which show differences in probabilities of evolving promoters, as well as the mechanisms underlying these evolutions. Furthermore, these mutations are well described and presented, showing the strength of emergent promoter motifs and their specific spacings from existing motifs within the sequence.

Impact of the work on the field, and the utility of the methods and data to the community:

This study enhances our understanding of the diverse mechanisms by which promoters can evolve or devolve, potentially improving models that predict mutational outcomes. While this study reveals complex mutational patterns, modeling them could significantly advance our ability to predict bacterial evolutionary trajectories and interpret genomes, bringing us closer to that goal.

Recent work in the field of bacterial gene regulation has raised interest in bidirectional promoter regions. While the authors do not discuss how mutations that raise expression in one direction may affect another, they have created an expansive dataset which may enable other groups to study this interesting phenomenon. Also, their variation of the Sort-Seq protocol will be a valuable example for other groups who may be interested in studying bidirectional expression. Lastly, this study may be of interests to groups studying eukaryotic regulation as it can inform how the evolution of transcription factor binding sites influences short-range interactions with local regulator elements.

Any additional context to understand the significance of the work:

Predicting whether a sequence drives promoter activity is a challenging task. By learning the types of mutations that create or destroy promoters, this study provides valuable insights for computational models aimed at predicting promoter activity.

Comments on revised version:

I am satisfied with the extensive changes made by the author. This manuscript is excellent.

I very much like the change in figures to incorporate the sequence information. It is great to see clear representations of the emergent sigma70 motifs and their spacing relative to existing motifs. This addition significantly improves the clarity of the findings.

The validation of mutations on a clean background is well-executed, and the results are convincing. I appreciate the effort put into validating their results. The additional analyses that include TGn and UP-element motifs are also well done and highly relevant, as these elements are known to compensate for weaker or absent -35 sequences.

Most or all perceived inconsistencies from the previous version have been resolved. While I don't think the fluorescence threshold of 1.5 a.u. for promoter activity is justified, the authors do acknowledge this shortcoming, and even empirically-derived thresholds are still technically arbitrary.

I particularly enjoyed Figure 1E, thank you for entertaining my analysis request! Also, the H-NS story is a nice addition showing how transcription factors influence this evolution

Overall, this revised manuscript is an excellent contribution to the field, and I have no further recommendations for improvement.

---

## [Referee Report · Reviewer #2 (Public review)]

Summary:

Fuqua et al investigated the relationship between prokaryotic box motifs and the activation of promoter activity using a mutagenesis sequencing approach. From generating thousands of mutant daughter sequences from both active and non-active promoter sequences they were able to produce a fantastic dataset to investigate potential mechanisms for promoter activation. From these large numbers of mutated sequences, they were able to generate mutual information with gene expression to identify key mutations relating to the activation of promoter island sequences.

Strengths:

The data generated from this paper is an important resource to address this question of promoter activation. Being able to link promoter modulated gene expression to mutational changes in previously nonactive promoter regions is exciting. This approach allows future large-scale studies to investigate evolutionary processes relating to changes in gene regulation in a statistically robust manner. Here there is a focus on the -10 and -35 boxes but other elements and interactions were explored including; H-NS binding, UP-element and TGn. Alongside this, the method of identifying key mutations using mutual information in this paper is well done and should be a standard in future studies for identifying regions of interest.

Weaknesses:

While the generation of the data is superb, as the authors have stated clearly themselves, there is a lot of scope for future studies to understand both causal relationships and utilise the data more effectively. The authors look at changes in regulatory expression based on a few observations that are treated independently but occur concurrently. While this study has backed up findings experimentally this may not always be possible. Previously this reviewer had suggested addressing this using complementary approaches such as analysis focusing on identifying important motifs, using something like a glm lasso regression to identify significant motifs, and then combining with mutational hotspot information would be more robust. The authors tried to implement such an approach in response to the review, but its complexity became beyond the scope. I look forward to the development of such methods that allow more complete exploration of similar datasets.

Comments on revised version:

The authors addressed all my previous comments. I believe the study is much improved and thank them for the time and effort they put into addressing the comments.

---

## [Referee Report · Reviewer #3 (Public review)]

This work brings a computational approach to the study of promoters and transcription. The paper is improved but there are still factual errors and implausible explanations. I am not convinced by the response from the authors, concerning the promoter -35 element, in their rebuttal.

Comments on author rebuttal:

- We respectfully but strongly disagree that our analysis has misrepresented the true nature of -35 boxes. First, accounting for more A's at position 5 in the PWM is not going to lead to a "critical error." This is because positions 4-6 of the motif barely have any information content (bits) compared to positions 1-3 (see Fig 1A).

The analysis does misrepresent the consensus -35 element, which is, unequivocally, TTGACA. I agree that positions 4-6 of the element are less well-conserved.

- This assertion is not just based on our own PWM, but based on ample precedent in the literature. In PMID 14529615, TTG is present in 38% of all -35 boxes, but ACA only in 8%.

This does not mean that TTGACA is not the consensus, or that "ACA" is not important at promoters where it's present.

- In PMID 29388765, with the -10 instance TATAAT, the -35 instance TTGCAA yields stronger promoters compared to the -35 instance TTGACA (See their Figure 3B).

This is a known phenomenon and results from "perfect" promoters being limited at the point of RNA polymerase promoter escape (because the RNAP struggles to "let go" of perfect promoters). This does not mean the TTGACA is not the consensus. Indeed, and this is a key point, it is evident in the figure the authors refer to that TTGACA stimulates more transcription than alternative -35 sequences when -10 elements are not perfect.

- In PMID 29745856 (Figure 2), the most information content lies in positions 1-3, with the A and C at position 5 both nearly equally represented, as in our PWM.

The motif shown in this paper suffers from exactly the same issue as the paper under review; the variable spacing between the -35 hexamer and -10 element isn't taken into account by MEME.

- In PMID 33958766 (Figure 1) an experimentally-derived -35 box is even reduced to a "partial" -35 box which only includes positions 1 and 2, with consensus: TTnnnn.

This paper does not show an "experimentally-derived -35 box" in Figure 1 (or anywhere else, as far as I can see).

- In addition, we did not derive the PWMs as the reviewer describes. The PWMs we use are based on computational predictions that are in excellent agreement with experimental results. Specifically, the PWMs we use are from PMID 29728462, which acquired 145 -10 and -35 box sequences from the top 3.3% of computationally predicted boxes from Regulon DB.

The paper mentioned states "for the genomic RNAP logo, sequences were taken from computationally predicted RNAP binding sites on RegulonDB" so these are not experimentally defined promoters? It's not obvious from the paper, or regulon DB, which sequences these are or how they were predicted.

- Thank you for pointing out that our original submission was incomplete in this regard. We address these concerns by new analyses, including some new experiments. First, Rho dependent termination is associated with the RUT motif, which is very rich in Cytosines (PMID: 30845912). Given that our sequences confer between 65%-78% of AT-content, canonical rho dependent termination is unlikely. However, we computationally searched for rho-dependent terminators using the available code from PMID: 30845912, but the algorithm did not identify any putative RUTs. Because this analysis was not informative, we did not include it in the paper.

I don't believe it is the case that Rho absolutely requires a RUT sequence. My understanding is that, if an RNA is not translated, Rho will intervene (e.g. see PMID: 18487194).

- We respectfully disagree that the reviewer's point is pertinent because what the reviewer is referring to is the likelihood that the sequence is a promoter, which indeed increases with AT content, but we are focused on the likelihood that a sequence becomes a promoter through DNA mutation

I disagree that this distinction is relevant. An AT-rich sequence will much more closely resemble a promoter by chance than a GC rich sequence. As an extreme example, the sequence TTTTTT can be converted into a reasonable -10 element by one change (to TATTTT) but the sequence GGGGGG can't.

---

## [Author Response]

The following is the authors’ response to the original reviews.

We performed multiple new experiments and analyses in response to the reviewers concerns, and incorporated the results of these analyses in the main text, and in multiple substantially revised or new figures. Before embarking on a point-by-point reply to the reviewers’ concerns, we here briefly summarize our most important revisions.

First, we addressed a concern shared by Reviewers #1-3 about a lack of information about our DNA sequences. To this end, we redesigned multiple figures (Figures 3, 4, 5, S8, S9, S10, S11, and S12) to include the DNA sequences of each tested promoter, the specific mutations that occurred in it, the resulting changes in position-weight-matrix (PWM) scores, and the spacing between promoter motifs. Second, Reviewers #1 and #2 raised concerns about a lack of validation of our computational predictions and the resulting incompleteness of the manuscript. To address this issue, we engineered 27 reporter constructs harboring specific mutations, and experimentally validated our computational predictions with them. Third, we expanded our analysis to study how a more complete repertoire of other sigma 70 promoter motifs such as the UP-element and the extended -10 / TGn motif affects gene expression driven by the promoters we study. Fourth, we addressed concerns by Reviewer #3 about the role of the Histone-like nucleoid-structuring protein (H-NS) in promoter emergence and evolution. We did this by performing both experiments and computational analyses, which are now shown in the newly added Figure 5. Fifth, to satisfy Reviewer #3’s concerns about missing details in the Discussion, we have rewritten this section, adding additional details and references.

We next describe these and many other changes in a point-by-point reply to each reviewer’s comments. In addition, we append a detailed list of changes to each section and figure to the end of this document.

**Reviewer #1 (Public Review):**
Summary:This study by Fuqua et al. studies the emergence of sigma70 promoters in bacterial genomes. While there have been several studies to explore how mutations lead to promoter activity, this is the first to explore this phenomenon in a wide variety of backgrounds, which notably contain a diverse assortment of local sigma70 motifs in variable configurations. By exploring how mutations affect promoter activity in such diverse backgrounds, they are able to identify a variety of anecdotal examples of gain/loss of promoter activity and propose several mechanisms for how these mutations interact within the local motif landscape. Ultimately, they show how different sequences have different probabilities of gaining/losing promoter activity and may do so through a variety of mechanisms.

We thank Reviewer #1 for taking the time to read and provide critical feedback on our manuscript. Their summary is fundamentally correct.

Major strengths and weaknesses of the methods and results:This study uses Sort-Seq to characterize promoter activity, which has been adopted by multiple groups and shown to be robust. Furthermore, they use a slightly altered protocol that allows measurements of bi-directional promoter activity. This combined with their pooling strategy allows them to characterize expressions of many different backgrounds in both directions in extremely high throughput which is impressive! A second key approach this study relies on is the identification of promoter motifs using position weight matrices (PWMs). While these methods are prone to false positives, the authors implement a systematic approach which is standard in the field. However, drawing these types of binary definitions (is this a motif? yes/no) should always come with the caveat that gene expression is a quantitative trait that we oversimplify when drawing boundaries.

The point is well-taken. To clarify this and other issues, we have added a section on the limitations of our work to the Discussion. Within this section we include the following sentences (lines 675-680):

“Additionally, future studies will be necessary to address the limitations of our own work. First, we use binary thresholding to determine (i) the presence or absence of a motif, (ii) whether a sequence has promoter activity or not, and (iii) whether a part of a sequence is a hotspot or not. While chosen systematically, the thresholds we use for these decisions may cause us to miss subtle but important aspects of promoter evolution and emergence.”

Their approach to randomly mutagenizing promoters allowed them to find many anecdotal examples of different types of evolutions that may occur to increase or decrease promoter activity. However, the lack of validation of these phenomena in more controlled backgrounds may require us to further scrutinize their results. That is, their explanations for why certain mutations lead or obviate promoter activity may be due to interactions with other elements in the 'messy' backgrounds, rather than what is proposed.

Thank you for raising this important point. To address it, we have conducted extensive new validation experiments for the newest version of this manuscript. For the “anecdotal” examples you described, we created 27 reporter constructs harboring the precise mutation that leads to the loss or gain of gene expression, and validated its ability to drive gene expression. The results from these experiments are in Figures 3, 4, 5, and Supplemental Figures S8-S11, and are labeled with a ′ (prime) symbol.

These experiments not only confirm the increases and decreases in fluorescence that our analysis had predicted. They also demonstrate, with the exception of two (out of 27) falsepositive discoveries, that background mutations do not confound our analysis. We mention these two exceptions (lines 364-367):

“In two of these hotspots, our validation experiments revealed no substantial difference in gene expression as a result of the hotspot mutation (Fig S8F′ and Fig S8J′). In both of these false positives, new -10 boxes emerge in locations without an upstream -35 box.”

An appraisal of whether the authors achieved their aims, and whether the results support their conclusions:The authors express a key finding that the specific landscape of promoter motifs in a sequence affects the likelihood that local mutations create or destroy regulatory elements. The authors have described many examples, including several that are non-obvious, and show convincingly that different sequence backgrounds have different probabilities for gaining or losing promoter activity. While this overarching conclusion is supported by the manuscript, the proposed mechanisms for explaining changes in promoter activity are not sufficiently validated to be taken for absolute truth. There is not sufficient description of the strength of emergent promoter motifs or their specific spacings from existing motifs within the sequence. Furthermore, they do not define a systematic process by which mutations are assigned to different categories (e.g. box shifting, tandem motifs, etc.) which may imply that the specific examples are assigned based on which is most convenient for the narrative.

To summarize, Reviewer #1 criticizes the following three aspects of our work in this comment. (1) The mechanisms we proposed are not sufficiently validated. (2) The description of motifs, spacing, and PWM scores are not shown. (3) How mutations are classified into different categories (i.e. box-shifting, tandem motifs, etc.) is not systematically defined.

These are all valid criticisms. In response, we performed an extensive set of follow-up experiments and analyses, and redesigned the majority of the figures. Here is a more detailed response to each criticism:

(1) Proposed mechanisms for explaining changes in promoter activity are not sufficiently validated. We engineered 27 reporter constructs harboring the specific mutations in the parents that we had predicted to change promoter activity. For each, we compared their fluorescence levels with their wild-type counterpart. The results from these experiments are in Figures 3 and 4, 5, and Supplemental Figures S8, S9, S10, S11, and S12, and are labeled with a ′ (prime) symbol.

(2) No sufficient description of the strength of emergent promoter motifs or their specific spacings. We redesigned the figures to include the DNA sequences of the parent sequences, as well as the degenerate consensus sequences for each mutation. We additionally now highlight the specific motif sequences, their respective PWM scores, and by how much the score changes upon mutation. Finally, we annotated the spacing of motifs. These changes are in Figures 3, 4, 5, and Supplemental Figures S8, S9, S10, S11, and S12.

We note that in many cases, high-scoring PWM hits for the same motif can overlap (i.e. two -10 motifs or two -35 motifs overlap). Additionally, the proximity of a -35 and -10 box does not guarantee that the two boxes are interacting. Together, these two facts can result in an ambiguity of the spacer size between two boxes. To avoid any reporting bias, we thus often report spacer sizes as a range (see Figure panels 4F, S8D, S8F-L, S9A, S9H, S10A, and S10E). The smallest spacer we annotate is in Figure 4F with 10 bp, and the largest is in Figure S8D with 26 bp. Any more “extreme” distances are not annotated and for the reader to decide if an interaction is present or not.

(3) No systematic process by which mutations are assigned to different categories such as box shifting, tandem motifs, etc. We opted to reformulate these categories completely, because the phenotypic effects of a previously mentioned “tandem motif” was actually a byproduct of H-NS repression (see the newly added Figure S12).

We also agree that the categories were ambiguous. We now introduce two terms: homo-gain and hetero-gain of -10 and -35 boxes. The manuscript now clearly defines these terms, and the relevant passage now reads as follows (lines 430-435):

“We found that these mutations frequently create new boxes overlapping those we had identified as part of a promoter

(Fig S9). This occurs when mutations create a -10 box overlapping a -10 box, a -35 box overlapping a -35 box, a -10 box overlapping a -35 box, or a -35 box overlapping a -10 box. We call the resulting event a “homo-gain” when the new box is of the same type as the one it overlaps, and otherwise a “hetero-gain”. In either case, the creation of the new box does not always destroy the original box.”

Impact of the work on the field, and the utility of the methods and data to the community: From this study, we are more aware of different types of ways promoters can evolve and devolve, but do not have a better ability to predict when mutations will lead to these effects. Recent work in the field of bacterial gene regulation has raised interest in bidirectional promoter regions. While the authors do not discuss how mutations that raise expression in one direction may affect another, they have created an expansive dataset that may enable other groups to study this interesting phenomenon. Also, their variation of the Sort-Seq protocol will be a valuable example for other groups who may be interested in studying bidirectional expression. Lastly, this study may be of interest to groups studying eukaryotic regulation as it can inform how the evolution of transcription factor binding sites influences short-range interactions with local regulator elements. Any additional context to understand the significance of the work:The task of computationally predicting whether a sequence drives promoter activity is difficult. By learning what types of mutations create or destroy promoters from this study, we are better equipped for this task.

We thank Reviewer #1 again for their time and their thoughtful comments.

**Reviewer #2 (Public Review):**
Summary:Fuqua et al investigated the relationship between prokaryotic box motifs and the activation of promoter activity using a mutagenesis sequencing approach. From generating thousands of mutant daughter sequences from both active and non-active promoter sequences they were able to produce a fantastic dataset to investigate potential mechanisms for promoter activation. From these large numbers of mutated sequences, they were able to generate mutual information with gene expression to identify key mutations relating to the activation of promoter island sequences.

We thank Reviewer #2 for reading and providing a thorough review of our manuscript.

Strengths:The data generated from this paper is an important resource to address this question of promoter activation. Being able to link the activation of gene expression to mutational changes in previously nonactive promoter regions is exciting and allows the potential to investigate evolutionary processes relating to gene regulation in a statistically robust manner. Alongside this, the method of identifying key mutations using mutual information in this paper is well done and should be standard in future studies for identifying regions of interest.

Thank you for your kind words.

Weaknesses:While the generation of the data is superb the focus only on these mutational hotspots removes a lot of the information available to the authors to generate robust conclusions. For instance.(1) The linear regression in S5 used to demonstrate that the number of mutational hotspots correlates with the likelihood of a mutation causing promoter activation is driven by three extreme points.

A fair criticism. In response, we have chosen to remove the analysis of this trend from the manuscript entirely. (Additionally, Pnew and mutual information calculations both relied on the fluorescence scores of daughter sequences, so the finding was circular in its logic.)

(2) Many of the arguments also rely on the number of mutational hotspots being located near box motifs. The context-dependent likelihood of this occurring is not taken into account given that these sequences are inherently box motif rich. So, something like an enrichment test to identify how likely these hot spots are to form in or next to motifs.

Another good point. To address it, we carried out a computational analysis where we randomly scrambled the nucleotides of each parent sequence while maintaining the coordinates for each mutual information “hotspot.” This scrambling results in significantly less overlap with hotspots and boxes. This analysis is now depicted in Figure 2C and described in lines 272-296.

(3) The link between changes in expression and mutations in surrounding motifs is assessed with two-sided Mann Whitney U tests. This method assumes that the sequence motifs are independent of one another, but the hotspots of interest occur either in 0, 3, 4, or 5s in sequences. There is therefore no sequence where these hotspots can be independent and the correlation causation argument for motif change on expression is weakened.

This is a fair criticism and a limitation of the MWU test. To better support our reasoning, we engineered 27 reporter constructs harboring the specific mutations in the parents that we had predicted to change promoter activity. For each, we compared their fluorescence levels with their wild-type counterpart. The results from these experiments are in Figures 3, 4, 5, and Supplemental Figures S8, S9, S10, S11, and S12 and are labeled with a ′ (prime) symbol.

These experiments not only confirm the increases and decreases in fluorescence that our analysis had predicted. They also demonstrate, with the exception of two (out of 27) falsepositive discoveries, that background mutations do not confound our analysis. We mention these two exceptions (lines 364-367):

“In two of these hotspots, our validation experiments revealed no substantial difference in gene expression as a result of the hotspot mutation (Fig S8F′ and Fig S8J′). In both of these false positives, new -10 boxes emerge in locations without an upstream -35 box.”

(4) The distance between -10 and -35 was mentioned briefly but not taken into account in the analysis.

We have now included these spacer distances where appropriate. These changes are in Figures 3, 4, 5, and Supplemental Figures S8, S9, S10, S11, and S12.

We note that in many cases, high-scoring PWM hits for the same motif can overlap (i.e. two -10 motifs or two -35 motifs overlap). Additionally, the proximity of a -35 and -10 box does not guarantee that the two boxes are interacting. Together, these two facts can result in an ambiguity of the spacer size between two boxes. To avoid any reporting bias, we thus often report spacer sizes as a range (see Figure panels 4F, S8D, S8F-L, S9A, S9H, S10A, and S10E). The smallest spacer we annotate is in Figure 4F with 10 bp, and the largest is in Figure S8D with 26 bp. More “extreme” distances are not annotated, and for the reader to decide if an interaction is present or not.

The authors propose mechanisms of promoter activation based on a few observations that are treated independently but occur concurrently. To address this using complementary approaches such as analysis focusing on identifying important motifs, using something like a glm lasso regression to identify significant motifs, and then combining with mutational hotspot information would be more robust.

This is a great idea, and we pursued it as part of the revision. For each parent sequence, we mapped the locations of all -10 and -35 box motifs in the daughters, then reduced each sequence to a binary representation, either encoding or not encoding these motifs, also referred to as a “hot-encoded matrix.” We subsequently performed a Lasso regression between the hot-encoded matrices and the fluorescence scores of each daughter sequence. The regression then outputs “weights” to each of the motifs in the daughters. The larger a motif’s weight is, the more the motif influences promoter activity. The Author response image 1 describes our workflow.

**Author response image 1. sa4fig1:** 

We really wanted this analysis to work, but unfortunately, the computational model does not act robustly, even when testing multiple values for the hyperparameter lambda (λ), which accounts for differences in model biases vs variance.

The regression assigns strong weights almost exclusively to -10 boxes, and assigns weak to even negative weights to -35 boxes. While initially exciting, these weights do not consistently align with the results from the 27 constructs with individual mutations that we tested experimentally. This ultimately suggests that the regression is overfitting the data.

We do think a LASSO-regression approach can be applied to explore how individual motifs contribute to promoter activity. However, effectively implementing such a method would require a substantially more complex analysis. We respectfully believe that such an approach would distract from the current narrative, and would be more appropriate for a computational journal in a future study.

Because this analysis was inconclusive, we have not made it part of the revised manuscript. However, we hope that our 27 experimentally validated new constructs with individual mutations are sufficient to address the reviewer’s concerns regarding independent verification of our computational predictions.

Other elements known to be involved in promoter activation including TGn or UP elements were not investigated or discussed.

Thank you for highlighting this potentially important oversight. In response, we have performed two independent analyses to explore the role of TGn in promoter emergence in evolution. First, we computationally searched for -10 boxes with the bases TGn immediately upstream of them in the parent sequences, and found 18 of these “extended -10 boxes” in the parents (lines 143145):

“On average, each parent sequence contains ~5.32 -10 boxes and ~7.04 -35 boxes (Fig S1). 18 of these -10 boxes also include the TGn motif upstream of the hexamer.”

However, only 20% of these boxes were found in parents with promoter activity (lines 182-185):

“We also note that 30% (15/50) of parents have the TGn motif upstream of a -10 box, but only 20% (3/15) of these parents have promoter activity (underlined with promoter activity: P4-RFP, P6-RFP, P8-RFP, P9-RFP, P10-RFP, P11GFP, P12-GFP, P17-GFP, P18-GFP, P18-RFP, P19-RFP, P22-RFP, P24-GFP, P25-GFP, P25-RFP). “

Second, we computationally searched through all of the daughter sequences to identify new -10 boxes with TGn immediately upstream. We found 114 -10 boxes with the bases TGn upstream. However, only 5 new -10 boxes (2 with TGn) were associated with increasing fluorescence (lines 338-345):

“On average, 39.5 and 39.4 new -10 and -35 boxes emerged at unique positions within the daughter sequences of each mutagenized parent (Fig 3A,B), with 1’562 and 1’576 new locations for -10 boxes and -35 boxes, respectively. ~22% (684/3’138) of these new boxes are spaced 15-20 bp away from their cognate box, and ~7.3% (114/1’562) of the new -10 boxes have the TGn motif upstream of them. However, only a mere five of the new -10 boxes and four of the new 35 boxes are significantly associated with increasing fluorescence by more than +0.5 a.u. (Fig 3C,D).”

In addition, we now study the role of UP elements. This analysis showed that the UP element plays a negligible role in promoter emergence within our dataset. It is discussed in a new subsection of the results (lines 591-608).

Collectively, these additional analyses suggest that the presence of TGn plus a -10 box is insufficient to create promoter activity, and that the UP element does not play a significant role in promoter emergence or evolution.

**Reviewer #3 (Public Review):**
Summary:Like many papers in the last 5-10 years, this work brings a computational approach to the study of promoters and transcription, but unfortunately disregards or misrepresents much of the existing literature and makes unwarranted claims of novelty. My main concerns with the current paper are outlined below although the problems are deeply embedded.

We thank Reviewer #3 for taking the time to review this manuscript. We have made extensive changes to address their concerns about our work.

Strengths:The data could be useful if interpreted properly, taking into account (i) the role of translation (ii) other promoter elements, and (iii) the relevant literature.Weaknesses:(1) Incorrect assumptions and oversimplification of promoters.- There is a critical error on line 68 and Figure 1A. It is well established that the -35 element consensus is TTGACA but the authors state TTGAAA, which is also the sequence represented by the sequence logo shown and so presumably the PWM used. It is essential that the authors use the correct -35 motif/PWM/consensus. Likely, the authors have made this mistake because they have looked at DNA sequence logos generated from promoter alignments anchored by either the position of the -10 element or transcription start site (TSS), most likely the latter. The distance between the TSS and -10 varies. Fewer than half of *E. coli* promoters have the optimal 7 bp separation with distances of 8, 6, and 5 bp not being uncommon (PMID: 35241653). Furthermore, the distance between the -10 and -35 elements is also variable (16,17, and 18 bp spacings are all frequently found, PMID: 6310517). This means that alignments, used to generate sequence logos, have misaligned -35 hexamers. Consequently, the true consensus is not represented. If the alignment discrepancies are corrected, the true consensus emerges. This problem seems to permeate the whole study since this obviously incorrect consensus/motif has been used throughout to identify sequences that resemble -35 hexamers.

We respectfully but strongly disagree that our analysis has misrepresented the true nature of -35 boxes. First, accounting for more A’s at position 5 in the PWM is not going to lead to a “critical error.” This is because positions 4-6 of the motif barely have any information content (bits) compared to positions 1-3 (see Fig 1A). This assertion is not just based on our own PWM, but based on ample precedent in the literature. In PMID 14529615, TTG is present in 38% of all -35 boxes, but ACA only in 8%. In PMID 29388765, with the -10 instance TATAAT, the -35 instance TTGCAA yields stronger promoters compared to the -35 instance TTGACA (See their Figure 3B).

In PMID 29745856 (Figure 2), the most information content lies in positions 1-3, with the A and C at position 5 both nearly equally represented, as in our PWM. In PMID 33958766 (Figure 1) an experimentally-derived -35 box is even reduced to a “partial” -35 box which only includes positions 1 and 2, with consensus: TTnnnn.

In addition, we did not derive the PWMs as the reviewer describes. The PWMs we use are based on computational predictions that are in excellent agreement with experimental results. Specifically, the PWMs we use are from PMID 29728462, which acquired 145 -10 and -35 box sequences from the top 3.3% of computationally predicted boxes from Regulon DB. See PMID 14529615 for the computational pipeline that was used to derive the PWMs, which independently aligns the -10 and -35 boxes to create the consensus sequences. The -35 PWMs significantly and strongly correlates with an experimentally derived -35 box (see Supporting Information from Figure S4 of Belliveau et al., PNAS 2017. Pearson correlation coefficient = 0.89). Within the 145 -35 boxes, the exact consensus sequence (TTGACA) that Reviewer #3 is concerned about is present 6 times in our matrix, and has a PWM score above the significance threshold. In other words, TTGACA, is classified to be a -35 box in our dataset.

We now provide DNA sequences for each of the figures to improve accessibility and reproducibility. A reader can now use any PWM or method they wish to interpret the data.

- An uninformed person reading this paper would be led to believe that prokaryotic promoters have only two sequence elements: the -10 and -35 hexamers. This is because the authors completely ignore the role of the TG motif, UP element, and spacer region sequence. All of these can compensate for the lack of a strong -35 hexamer and it's known that appending such elements to a lone -10 sequence can create an active promoter (e.g. PMIDs 15118087, 21398630, 12907708, 16626282, 32297955). Very likely, some of the mutations, classified as not corresponding to a -10 or -35 element in Figure 2, target some of these other promoter motifs.

Thank you for bringing this oversight to our attention. We have performed two independent analyses to explore the role of TGn in promoter emergence in evolution. First, we computationally searched for -10 boxes with the bases TGn immediately upstream of them in the parent sequences, and found 18 of these “extended -10 boxes” in the parents (lines 143145):

“On average, each parent sequence contains ~5.32 -10 boxes and ~7.04 -35 boxes (Fig S1). 18 of these -10 boxes also include the TGn motif upstream of the hexamer.”

However, only 20% of these boxes were found in parents with promoter activity (lines 182-185):

“We also note that 30% (15/50) of parents have the TGn motif upstream of a -10 box, but only 20% (3/15) of these parents have promoter activity (underlined with promoter activity: P4-RFP, P6-RFP, P8-RFP, P9-RFP, P10-RFP, P11GFP, P12-GFP, P17-GFP, P18-GFP, P18-RFP, P19-RFP, P22-RFP, P24-GFP, P25-GFP, P25-RFP).”

Second, we computationally searched through all of the daughter sequences to identify new -10 boxes with TGn immediately upstream. We found 114 -10 boxes with the bases TGn upstream. However, only 5 new -10 boxes (2 with TGn) were associated with increasing fluorescence (lines 338-345):

“On average, 39.5 and 39.4 new -10 and -35 boxes emerged at unique positions within the daughter sequences of each mutagenized parent (Fig 3A,B), with 1’562 and 1’576 new locations for -10 boxes and -35 boxes, respectively. ~22% (684/3’138) of these new boxes are spaced 15-20 bp away from their cognate box, and ~7.3% (114/1’562) of the new -10 boxes have the TGn motif upstream of them. However, only a mere five of the new -10 boxes and four of the new 35 boxes are significantly associated with increasing fluorescence by more than +0.5 a.u. (Fig 3C,D).”

In addition, we now study the role of UP elements. This analysis showed that the UP element plays a negligible role in promoter emergence within our dataset. It is discussed in a new subsection of the results (lines 591-608) and in the newly added Figure S13.

Collectively, these additional analyses suggest that the presence of TGn plus a -10 box is insufficient to create promoter activity, and that the UP element does not play a significant role in promoter emergence or evolution.

- The model in Figure 4C is highly unlikely. There is no evidence in the literature that RNAP can hang on with one "arm" in this way. In particular, structural work has shown that sequencespecific interactions with the -10 element can only occur after the DNA has been unwound (PMID: 22136875). Further, -10 elements alone, even if a perfect match to the consensus, are non-functional for transcription. This is because RNAP needs to be directed to the -10 by other promoter elements, or transcription factors. Only once correctly positioned, can RNAP stabilise DNA opening and make sequence-specific contacts with the -10 hexamer. This makes the notion that RNAP may interact with the -10 alone, using only domain 2 of sigma, extremely unlikely.

This is a valid criticism, and we thank the reviewer for catching this problem. In response, we have removed the model and pertinent figures throughout the entire manuscript.

(2) Reinventing the language used to describe promoters and binding sites for regulators.- The authors needlessly complicate the narrative by using non-standard language. For example, On page 1 they define a motif as "a DNA sequence computationally predicted to be compatible with TF binding". They distinguish this from a binding site "because binding sites refer to a location where a TF binds the genome, rather than a DNA sequence". First, these definitions are needlessly complicated, why not just say "putative binding sites" and "known binding sites" respectively? Second, there is an obvious problem with the definitions; many "motifs" with also be "bindings sites". In fact, by the time the authors state their definitions, they have already fallen foul of this conflation; in the prior paragraph they stated: "controlled by DNA sequences that encode motifs for TFs to bind". The same issue reappears throughout the paper.

We agree that this was needlessly complicated. We now just refer to every sequence we study as a motif. A -10 box is a motif, a -35 box is a motif, a putative H-NS binding site is an H-NS motif, etc. The word “binding site” no longer occurs in the manuscript.

- The authors also use the terms "regulatory" and non-regulatory" DNA. These terms are not defined by the authors and make little sense. For instance, I assume the authors would describe promoter islands lacking transcriptional activity (itself an incorrect assumption, see below)as non-regulatory. However, as horizontally acquired sections of AT-rich DNA these will all be bound by H-NS and subject to gene silencing, both promoters for mRNA synthesis and spurious promoters inside genes that create untranslated RNAs. Hence, regulation is occurring.

Another fair point. We have thus changed the terminology throughout to “promoter” and “nonpromoter.”

- Line 63: "In prokaryotes, the primary regulatory sequences are called promoters". Promoters are not generally considered regulatory. Rather, it is adjacent or overlapping sites for TFs that are regulatory. There is a good discussion of the topic here (PMID: 32665585).

We have rewritten this. The sentence now reads (lines 67-69):

“A canonical prokaryotic promoter recruits the RNA polymerase subunit σ70 to transcribe downstream sequences (Burgess et al., 1969; Huerta and Collado-Vides, 2003; Paget and Helmann, 2003; van Hijum et al., 2009).”

(3) The authors ignore the role of translation.- The authors' assay does not measure promoter activity alone, this can only be tested by measuring the amount of RNA produced. Rather, the assay used measures the combined outputs of transcription and translation. If the DNA fragments they have cloned contain promoters with no appropriately positioned Shine-Dalgarno sequence then the authors will not detect GFP or RFP production, even though the promoter could be making an RNA (likely to be prematurely terminated by Rho, due to a lack of translation). This is known for promoters in promoter islands (e.g. Figure 1 in PMID: 33958766).

We agree that this is definitely a limitation of our study, which we had not discussed sufficiently. In response, we now discuss this limitation in a new section of the discussion (lines 680-686):

“Second, we measure protein expression through fluorescence as a readout for promoter activity. This readout combines transcription and translation. This means that we cannot differentiate between transcriptional and post-transcriptional regulation, including phenomena such as premature RNA termination (Song et al., 2022; Uptain and Chamberlin, 1997), post-transcriptional modifications (Mohanty and Kushner, 2006), and RNA-folding from riboswitch-like sequences (Mandal and Breaker, 2004).”

- In Figure S6 it appears that the is a strong bias for mutations resulting in RFP expression to be close to the 3' end of the fragment. Very likely, this occurs because this places the promoter closer to RFP and there are fewer opportunities for premature termination by Rho.

The reviewer raises a very interesting possibility. To validate it, we have performed the following analysis. We took the RFP expression values from the 9’934 daughters with single mutations in all 25 parent sequences (P1-RFP, P2-RFP, … P25-RFP), and plotted the location of the single mutation (horizontal axis) against RFP expression (vertical axis) in Author response image 2.

**Author response image 2. sa4fig2:** 

The distribution is uniform across the sequences, showing that distance from the RBS is not likely the reason for this observation. Since this analysis was uninformative with respect to distance from the RBS, we chose not to include it in the manuscript.

(4) Ignoring or misrepresenting the literature.- As eluded to above, promoter islands are large sections of horizontally acquired, high ATcontent, DNA. It is well known that such sequences are (i) packed with promoters driving the expression on RNAs that aren't translated (ii) silenced, albeit incompletely, by H-NS and (iii) targeted by Rho which terminates untranslated RNA synthesis (PMIDs: 24449106, 28067866, 18487194). None of this is taken into account anywhere in the paper and it is highly likely that most, if not all, of the DNA sequences the authors have used contain promoters generating untranslated RNAs.

Thank you for pointing out that our original submission was incomplete in this regard. We address these concerns by new analyses, including some new experiments. First, Rhodependent termination is associated with the RUT motif, which is very rich in Cytosines (PMID: 30845912). Given that our sequences confer between 65%-78% of AT-content, canonical rhodependent termination is unlikely. However, we computationally searched for rho-dependent terminators using the available code from PMID: 30845912, but the algorithm did not identify any putative RUTs. Because this analysis was not informative, we did not include it in the paper.

We analyzed the role of H-NS on promoter emergence and evolution within our dataset using both experimental and computational approaches. These additional analyses are now shown in the newly-added Figure 5 and the newly-added Figure S12. We found that H-NS represses P22-GFP and P12-RFP and affects the bidirectionality of P20. More specifically, to analyze the effects of H-NS, we first compared the fluorescence levels of parent sequences in a Δhns background vs the wild-type (dh5α) background in Figure 5A. We found 6 candidate H-NS targets, with P22-GFP and P12-RFP exhibiting the largest changes in fluorescence (lines 496506):

“We plot the fluorescence changes in Fig 5A as distributions for the 50 parents, where positive and negative values correspond to an increase or decrease in fluorescence in the Δhns background, respectively. Based on the null hypothesis that the parents are not regulated by H-NS, we classified outliers in these distributions (1.5 × the interquartile range) as H-NS-target candidates. We refer to these outliers as “candidates” because the fluorescence changes could also result from indirect trans-effects from the knockout (Mattioli et al., 2020; Metzger et al., 2016). This approach identified 6 candidates for H-NS targets (P2-GFP, P19-GFP, P20-GFP, P22-GFP, P12-RFP, and P20-RFP). For GFP, the largest change occurs in P22-GFP, increasing fluorescence ~1.6-fold in the mutant background (two-tailed t-test, p=1.16×10-8) (Fig 5B). For RFP, the largest change occurs in P12-RFP, increasing fluorescence ~0.5-fold in the mutant background (two-tailed t-test, p=4.33×10-10) (Fig 5B).”

We also observed that the Δhns background affected the bidirectionality of P20 (lines 507-511):

“We note that for template P20, which is a bidirectional promoter, GFP expression increases ~2.6-fold in the Δhns background (two-tailed t-test, p=1.59×10-6). Simultaneously, RFP expression decreases ~0.42-fold in the Δhns background (two-tailed t-test, p=4.77×10-4) (Fig S12A). These findings suggest that H-NS also modulates the directionality of P20’s bidirectional promoter through either cis- or trans-effects.”

We then searched for regions where losing H-NS motifs in hotspots significantly changed fluorescence. We identified 3 motifs in P12-RFP and P22-GFP (lines 522-528):

“For P22-GFP, a H-NS motif lies 77 bp upstream of the mapped promoter. Mutations which destroy this motif significantly increase fluorescence by +0.52 a.u. (two-tailed MWU test, q=1.07×10-3) (Fig 5E). For P12-RFP, one H-NS motif lies upstream of the mapped promoter’s -35 box, and the other upstream of the mapped promoter’s -10 box. Mutations that destroy these H-NS motifs significantly increase fluorescence by +0.53 and +0.51 a.u., respectively (two-tailed MWU test, q=3.28×10-40 and q=4.42 ×10-50) (Fig 5F,G). Based on these findings, we conclude that these motifs are bound by H-NS.”

We are grateful for the suggestion to look at the role of H-NS in our dataset. Our analysis revealed a more plausible explanation to what we formerly referred to as a “Tandem Motif” in the original submission. Previously, we had shown that in P12-RFP, when a -35 box is created next to the promoter’s -35 box, or a -10 box next to the promoter’s -10 box, that expression decreases. These new -10 and -35 boxes, however, also overlap with the two H-NS motifs in P12-RFP. We tested these exact point mutations in reporter plasmids and in the Δhns background, and found that the Δhns background rescues this loss in expression (see Figure S12). This analysis is in the newly added subsection: “The binding of H-NS changes when new 10 and -35 boxes are gained” and can be found at lines 529-563. We summarize the findings in a final paragraph of the section (lines 556-563):

“To summarize, we present evidence that H-NS represses both P22-GFP and P12-RFP in cis. H-NS also modulates the bidirectionality of P20-GFP/RFP in cis or trans. In P22-GFP, the strongest H-NS motif lies upstream of the promoter. In P12-RFP, the strongest H-NS motifs lie upstream of the -10 and -35 boxes of the promoter. We note that there are 16 additional H-NS motifs surrounding the promoter in P12-RFP that may also regulate P12-RFP (Fig S12G). Mutations in two of these two H-NS motifs can create additional -10 and -35 boxes that appear to lower expression. However, the effects of these mutations are insignificant in the absence of H-NS, suggesting that these mutations actually modulate H-NS binding.”

We also agree that the majority of these sequences are likely driving the expression of many untranslated RNAs (see Purtov et al., 2014). We thus now define a promoter more carefully as follows (lines 113-119):

“In this study, we define a promoter as a DNA sequence that drives the expression of a (fluorescent) protein whose expression level, measured by its fluorescence, is greater than a defined threshold. We use a threshold of 1.5 arbitrary units (a.u.) of fluorescence. This definition does not distinguish between transcription and translation. We chose it because protein expression is usually more important than RNA expression whenever natural selection acts on gene expression, because it is the primary phenotype visible to natural selection (Jiang et al., 2023).”

We also state this as a limitation of our study in the Discussion (lines 680-686):

“Second, we measure protein expression through fluorescence as a readout for promoter activity. This readout combines transcription and translation. This means that we cannot differentiate between transcriptional and post-transcriptional regulation, including phenomena such as premature RNA termination (Song et al., 2022; Uptain and Chamberlin, 1997), post-transcriptional modifications (Mohanty and Kushner, 2006), and RNA-folding from riboswitch-like sequences (Mandal and Breaker, 2004).”

- The authors state that GC content does not correlate with the emergence of new promoters. It is known that GC content does correlate to the emergence of new promoters because promoters are themselves AT-rich DNA sequences (e.g. see Figure 1 of PMID: 32297955). There are two reasons the authors see no correlation in this work. First, the DNA sequences they have used are already very AT-rich (between 65 % and 78 % AT-content). Second, they have only examined a small range of different AT-content DNA (i.e. between 65 % and 78 %). The effect of AT-content on promoter emerge is most clearly seen between AT-content of between around 40 % and 60 %. Above that level, the strong positive correlation plateaus.

We respectfully disagree that the reviewer’s point is pertinent because what the reviewer is referring to is the likelihood that the sequence is a promoter, which indeed increases with AT content, but we are focused on the likelihood that a sequence becomes a promoter through DNA mutation. We note that if a DNA sequence is more AT-rich, then it is more likely to have -10 and -35 boxes, because their consensus sequences are also AT-rich. However, H-NS and other transcriptional repressors also bind to AT-rich sequences. This could also explain the saturation observed above 60% AT-content in PMID 32297955. Perhaps we can address this trend in future works.

- Once these authors better include and connect their results to the previous literature, they can also add some discussion of how previous papers in recent years may have also missed some of this important context.

We apologize for this oversight. We have rewritten the Discussion section to include the following points below. Many of the newly added references come from the group of David Grainger, who works on H-NS repression, bidirectional promoters, promoter emergence, promoter motifs, and spurious transcription in *E. coli*. More specifically:

(1) The role of pervasive transcription and the likelihood of promoter emergence (lines 614-621):

“Instead, we present evidence that promoter emergence is best predicted by the level of background transcription each non-promoter parent produces, a phenomenon also referred to as “pervasive transcription” (Kapranov et al., 2007).

From an evolutionary perspective, this would suggest that sequences that produce such pervasive transcripts – including the promoter islands (Panyukov and Ozoline, 2013) and the antisense strand of existing promoters (Dornenburg et al., 2010; Warman et al., 2021), may have a proclivity for evolving de-novo promoters compared to other sequences (Kapranov et al., 2007; Wade and Grainger, 2014).”

(2) How our results contradict the findings from Bykov et al., 2020 (lines 622-640):

“A previous study randomly mutagenized the appY promoter island upstream of a GFP reporter, and isolated variants with increased and decreased GFP expression. The authors found that variants with higher GFP expression acquired mutations that (1) improve a -10 box to better match its consensus, and simultaneously (2) destroy other -10 and -35 boxes (Bykov et al., 2020). The authors concluded that additional -10 and -35 boxes repress expression driven by promoter islands. Our data challenge this conclusion in several ways.

First, we find that only ~13% of -10 and -35 boxes in promoter islands actually contribute to promoter activity. Extrapolating this percentage to the appY promoter island, ~87% (100% - 13%) of the motifs would not be contributing to its activity. Assuming the appY promoter island is not an outlier, this would insinuate that during random mutagenesis, these inert motifs might have accumulated mutations that do not change fluorescence. Indeed, Bykov et al. (Bykov et al., 2020) also found that a similar frequency of -10 and -35 boxes were destroyed in variants selected for lower GFP expression, which supports this argument. Second, we find no evidence that creating a -10 or -35 box lowers promoter activity in any of our 50 parent sequences. Third, we also find no evidence that destruction of a -10 or -35 box increases promoter activity without plausible alternative explanations, i.e. overlap of the destroyed box with a H-NS site, destruction of the promoter, or simultaneous creation of another motif as a result of the destruction. In sum, -10 and 35 boxes are not likely to repress promoter activity.”

(3) How other sequence features besides the -10 and -35 boxes may influence promoter emergence and activity (lines 661-671):

“These findings suggest that we are still underestimating the complexity of promoters. For instance, the -10 and -35 boxes, extended -10, and the UP-element may be one of many components underlying promoter architecture. Other components may include flanking sequences (Mitchell et al., 2003), which have been observed to play an important role in eukaryotic transcriptional regulation (Afek et al., 2014; Chiu et al., 2022; Farley et al., 2015; Gordân et al., 2013). Recent studies on *E. coli* promoters even characterize an AT-rich motif within the spacer sequence (Warman et al., 2020), and other studies use longer -10 and -35 box consensus sequences (Lagator et al., 2022). Another possibility is that there is much more transcriptional repression in the genome than anticipated (Singh et al., 2014). This would also coincide with the observed repression of H-NS in P22-GFP and P12-RFP, and accounts of H-NSrepression in the full promoter island sequences (Purtov et al., 2014).”

(4) The limits of our experimental methodology (lines 675-686):

“Additionally, future studies will be necessary to address the limitations of our own work. First, we use binary thresholding to determine (i) the presence or absence of a motif, (ii) whether a sequence has promoter activity or not, and (iii) whether a part of a sequence is a hotspot or not. While chosen systematically, the thresholds we use for these decisions may cause us to miss subtle but important aspects of promoter evolution and emergence. Second, we measure protein expression through fluorescence as a readout for promoter activity. This readout combines transcription and translation. This means that we cannot differentiate between transcriptional and post-transcriptional regulation, including phenomena such as premature RNA termination (Song et al., 2022; Uptain and Chamberlin, 1997), posttranscriptional modifications (Mohanty and Kushner, 2006), and RNA-folding from riboswitch-like sequences (Mandal and Breaker, 2004) “

(5) An updated take-home message (lines 687-694):

“Overall, our study demonstrates that -10 and -35 boxes neither prevent existing promoters from driving expression, nor do they prevent new promoters from emerging by mutation. It shows how mutations can create new -10 and -35 boxes near or on top of preexisting ones to modulate expression. However, randomly creating a new -10 or -35 box will rarely create a new promoter, even if the new box is appropriately spaced upstream or downstream of a cognate box. Ultimately our study demonstrates that promoter models need to be further scrutinized, and that using mutagenesis to create de-novo promoters can provide new insights into promoter regulatory logic.”

(5) Lack of information about sequences used and mutations.- To properly assess the work any reader will need access to the sequences cloned at the start of the work, where known TSSs are within these sequences (ideally +/- H-NS, which will silence transcription in the chromosomal context but may not when the sequences are removed from their natural context and placed in a plasmid). Without this information, it is impossible to assess the validity of the authors' work.

Thank you for raising this point. Please see Data S1 for the 25 template sequences (P1-P25) used in this study, and Data S2 for all of the daughter sequences.

For brevity, we have addressed the reviewer’s request to look at the role of H-NS in their comment (4) “Ignoring or misrepresenting the literature.”

We do not have information about the predicted transcription start sites (TSS) for the parent sequences because the program which identified them (Platprom) is no longer available. Regardless, having TSS coordinates would not validate or invalidate our findings, since we already know that the promoter islands produce short transcripts throughout their sequences, and we are primarily interested in promoters which can produce complete transcripts.

- The authors do not account for the possibility that DNA sequences in the plasmid, on either side of the cloned DNA fragment, could resemble promoter elements. If this is the case, then mutations in the cloned DNA will create promoters by "pairing up" with the plasmid sequences. There is insufficient information about the DNA sequences cloned, the mutations identified, or the plasmid, to determine if this is the case. It is possible that this also accounts for mutational hotspots described in the paper.

We agree that these are important points. To address the criticism that we provided insufficient information, we now redesigned all our figures to provide this information. Specifically, the figures now include the DNA sequences, their PWM predictions, and the exact mutations that lead to promoter activity. The figures with these changes are Figures 3, 4, 5, and Supplemental Figures S8, S9, S10, S11, and S12. We now also provide more details about pMR1 in a new section of the methods (lines 740-748):

“Plasmid MR1 (pMR1)

The plasmid MR1 (pMR1) is a variant of the plasmid RV2 (pRV2) in which the kan resistance gene has been swapped with the cm resistance gene (Guazzaroni and Silva-Rocha, 2014). Plasmid pMR1 encodes the BBa_J34801 ribosomal binding site (RBS, AAAGAGGAGAAA) 6 bp upstream of the start codon for GFP(LVA). The plasmid also encodes a putative RBS (AAGGGAGG) (Cazemier et al., 1999) 5 bp upstream of the start codon for mCherry on the opposite strand.

The plasmid additionally contains the low-to-medium copy number origin of replication p15A (Westmann et al., 2018).

A map of the plasmid is available on the Github repository: https://github.com/tfuqua95/promoter_islands “

The reviewer also makes a valid point about promoter elements of the plasmid itself. We addressed it with the following new analyses. First we re-examined each of the examples where new -10 and -35 boxes are gained or lost, to see if any of these hotspots occur on the flanking ends of the parent sequences. We looked specifically at the ends because they could potentially interact with -10 and -35 box-like sequences on the plasmid to form a promoter.

Only one of these hotspots (out of 27) occurred at the end of the cloned sequences, and is thus a candidate for the phenomenon the reviewer hypothesized. This hotspot occurs in P9-GFP, where gaining a -10 box at the left flank increases expression (see Figure S8E-F’). There is indeed a -35 box 22-23 bp upstream of this -10 box on the plasmid, which could potentially affect promoter activity.

We tested the GFP expression of a construct harboring the point mutation which creates this -10 box on the left flank of P9-GFP. However, there was no significant difference in fluorescence between this construct and the wile-type P9-GFP (see Figure S8E-F’). Thus, this -35 box on pMR1 is not likely creating a new promoter.

(6) Overselling the conclusions.Line 420: The paper claims to have generated important new insights into promoters. At the same time, the main conclusion is that "Our study demonstrates that mutations to -10 and -35 boxes motifs are the primary paths to create new promoters and to modulate the activity of existing promoters". This isn't new or unexpected. People have been doing experiments showing this for decades. Of course, mutations that make or destroy promoter elements create and destroy promoters. How could it be any other way?

In hindsight, we agree that the original conclusion was not very novel. Our new conclusion is that -10 and -35 boxes do not repress transcription, and that our current promoter models, even with the additional motifs like the UP-element and the extended -10, are insufficient to understand promoters (lines 687-694):

“Overall, our study demonstrates that -10 and -35 boxes neither prevent existing promoters from driving expression, nor do they prevent new promoters from emerging by mutation. It shows how mutations can create new -10 and -35 boxes near or on top of preexisting ones to modulate expression. However, randomly creating a new -10 or -35 box will rarely create a new promoter, even if the new box is appropriately spaced upstream or downstream of a cognate box. Ultimately our study demonstrates that promoter models need to be further scrutinized, and that using mutagenesis to create de-novo promoters can provide new insights into promoter regulatory logic.”

**Recommendations for the authors:**

**Reviewer #1 (Recommendations For The Authors):**
I would like to start by thanking the authors for presenting an interesting and well-written article for review. This paper is a welcome addition to the field, addressing modern questions in the longstanding area of bacterial gene regulation. It is both enlightening and inspiring. While I do have suggestions, I hope these are not perceived as a lack of optimism for the work.

Thank you for your kind words and suggestions, and for providing an astute and constructive review. We feel that manuscript has greatly improved with your suggested changes.

ABSTRACT:Line 11: The sentence, "It is possible that these motifs influence..." Could be rewritten to be clearer as it is the most important point of the manuscript. It is not obvious that you're talking about how the local landscape of motifs affects the probability of promoters evolving/devolving in this location.

We have changed the sentence to read, “Here, we ask whether the presence of such motifs in different genetic sequences influences promoter evolution and emergence.”

INTRODUCTION:Line 68: Is the -35 consensus motif not TTGACA? Here it is listed as TTGAAA.

Corrected from TTGAAA to TTGACA

RESULTS:Line 92-94. In finding that the. The main takeaway from this work is that different sequences have different likelihoods of mutations creating promoters and so I believe this claim could be explored deeper with more quantitative information. Could the authors supplement this claim by including? Could you look at whether there is a correlation between the baseline expression of a parent sequence and Pnew? I expect even the inactive sequences to have some variability in measured expression.

Thank you for this great idea. We followed up on it by plotting the baseline parent sequence fluorescence scores against Pnew. You are indeed correct, i.e., Pnew increases with baseline expression following a sigmoid function, and is now shown in Figure 1D. To report our new observations, we have added the following section to the Results (lines 219-232):

“Although mutating each of the 40 non-promoter parent sequences could create promoter activity, the likelihood Pnew that a mutant has promoter activity, varies dramatically among parents. For each non-promoter parent, Fig 1D shows the percentage of active daughter sequences. The median Pnew is 0.046 (std. ± 0.078), meaning that ~4.6% of all mutants have promoter activity. The lowest Pnew is 0.002 (P25-GFP) and the highest 0.41 (P8-RFP), a 205-fold difference.

We hypothesized that these large differences in Pnew could be explained by minute differences in the fluorescence scores of each parent, particularly if its score was below 1.5 a.u. Plotting the fluorescence scores of each parent (N=50) and their respective Pnew values as a scatterplot (Fig 1E), we can fit these values to a sigmoid curve (see methods). This finding helps to explain why P8-RFP has a high Pnew (0.41) and P25-GFP a low Pnew (0.002), as their fluorescence scores are 1.380 and 1.009 a.u., respectively. The fact that the inflection point of the fitted curve is at 1.51 a.u. further justifies our use of 1.5 a.u. as a cutoff for promoter and non-promoter activity.”

Another potentially interesting analysis would be to see if k-mer content is correlated with Pnew. That is, determine the abundance of all hexamers in the sequence and see if Pnew is correlated with the number of hexamers present that is one nucleotide distance away from the consensus motifs (such as TcGACA or TAcAAT).

We performed the suggested analysis by searching for k-mers that correlate with Pnew and found that no k-mer significantly correlates with Pnew (lines 240-248):

“We then asked whether any k-mers ranging from 1-6 bp correlated with the non-promoter Pnew values (5,460 possible k-mers). 718 of these 1-6 bp k-mers are present 3 or more times in at least one non-promoter parent. We calculated a linear regression between the frequency of these 718 k-mers and each Pnew value, and adjusted the p-values to respective q-values (Benjamini-Hochberg correction, FDR=0.05). This analysis revealed six k-mers: CTTC, GTTG,

ACTTC, GTTGA, AACTTC, TAACTT which correlate with Pnew. However, these correlations are heavily influenced by an outlying Pnew value of 0.41 (P8-RFP) (Fig S5C-H), and upon removing P8-RFP from the analysis, no k-mer significantly correlates with Pnew (data not shown)”

Line 152-157: How did you define the thresholds for 'active' or 'inactive'? It is not clear in the methods how this distinction was made.

We have more clearly defined these thresholds in the text. A sequence with promoter activity has a fluorescence score greater than 1.5 a.u. (lines 168-172):

“We declared a daughter sequence to have promoter activity or to be a promoter if its score was greater than or equal to 1.5 a.u., as this score lies at the boundary between no fluorescence and weak fluorescence based on the sort-seq bins (methods). Otherwise, we refer to a daughter sequence as having no promoter activity or being a non-promoter.”

Lines: 152-157: In trying to find the parent expression levels, no figure was available showing the distribution of parent expression levels. Furthermore, In looking at Data S2 & filtering out for sequences with distance 0 from the parent, I found the most active sequences did not match up with the sequences described as active in this section (e.g. p19 and p20 have a higher topstrand mean over P22, yet are not listed as active top strand sequences).

We really appreciate you taking the time to examine the supplemental data. We previously listed the parents that had *only* GFP activity but no RFP activity (P22), and *only* RFP activity but no GFP activity (P6, P12, P13, P18, P21). We then said that P19 and P20 were bidirectional promoters, because they showed *both* GFP and RFP activity. In hindsight, we realize that our wording was confusing. We thus rewrote the affected paragraph, such that the bidirectional promoters are now in both lists of GFP/RFP active parents. We also now make the distinction between “templates” which comprise our 25 promoter island fragments, and “parents”, where we treat both strands separately (50 parents total). The paragraph in question now reads (lines 173-187):

“Because some sequences in our library are unmutated parent sequences, we determined that 10/50 of the parent sequences already encode promoter activity before mutagenesis. Specifically, three parents drove expression on the top strand (P19-GFP, P20-GFP, P22-GFP), and five did on the bottom strand (P6-RFP, P12-RFP, P13-RFP, P18-RFP, P19-RFP, P20-RFP, P21-RFP). Two parents harbor bidirectional promoters (P19 and P20). The remaining 40 parent sequences are non-promoters, with an average fluorescence score of 1.39 a.u. We note that some of these parents have a fluorescence score higher than 1.39 a.u., but less than 1.50 a.u. such as P8-RFP (1.38 a.u.), P16-RFP (1.39 a.u.), P9-GFP (1.49 a.u.), and P1-GFP (1.47 a.u.). Whether these are truly “promoters” or not, is based solely on our threshold value of 1.5 a.u. We also note that 30% (15/50) of parents have the TGn motif upstream of a -10 box, but only 20% (3/15) of these parents have promoter activity (underlined with promoter activity: P4-RFP, P6-RFP, P8-RFP, P9RFP, P10-RFP, P11-GFP, P12-GFP, P17-GFP, P18-GFP, P18-RFP, P19-RFP, P22-RFP, P24-GFP, P25-GFP, P25RFP). See Fig S4 for fluorescence score distributions for each parent and its daughters, and Data S2 for all daughter sequence fluorescence scores.”

Please include a supplementary figure showing the different parent expression levels (GFP mean +/- sd). Also, please explain the discrepancy in the 'active sequences' compared to Data S2 or correct my misunderstanding.

We have added this plot to Figure S4B. The discrepancy arose because we listed the parents that had *only* GFP activity but no RFP activity (P22), and *only* RFP activity but no GFP activity (P6, P12, P13, P18, P21). We then said that P19 and P20 were bidirectional promoters, because they showed *both* GFP and RFP activity. previous response regarding the ambiguity.

Line 182: I do not see 'Fuqua and Wagner 2023' in the references (though I am familiar with the preprint).

We have added Fuqua and Wagner, BiorXiv 2023 to the references.

Lines 197 - 200: The distribution of hotspot locations should be compared to the distribution of mutations in the library. e.g. It is not notable that 17% of mutations are in -10 motifs if 17% of all mutations are in -10 motifs.

Thank you for raising this point. To address it, we carried out a computational analysis where we randomly scrambled the nucleotides of each parent sequence while maintaining the coordinates for each mutual information “hotspot.” This scrambling results in significantly less overlap with hotspots and boxes. This analysis is now depicted in Figure 2C and written in lines 272-296.

Lines 253-264: Examples 3B, 3D, and 3F should indicate the spacing between the new and existing motifs. Are these close to the 15-19 bp spacer lengths preferred by sigma70?

Point well taken. We now annotate the spacing of motifs in Figures 3, 4, 5, and Supplemental Figures S8, S9, S10, and S11. We note that in many cases, high-scoring PWM hits for the same motif can overlap (i.e. two -10 motifs or two -35 motifs overlap). Additionally, the proximity of a 35 and -10 box does not guarantee that the two boxes are interacting. Together, these two facts can result in an ambiguity of the spacer size between two boxes. To avoid any reporting bias, we thus often report spacer sizes as a range (see Figure panels 4F, S8D, S8F-L, S9A, S9H, S10A, and S10E). The smallest spacer we annotate is in Figure 4F with 10 bp, and the largest is in Figure S8D with 26 bp. Any more “extreme” distances are not annotated, and for the reader to decide if an interaction is present or not.

Line 255: While fun, I am concerned about the 'Shiko' analogy. My understanding is the prevailing theory is that -35 recognition occurs before -10 recognition (https://doi.org/10.1073/pnas.94.17.9022, 10.1101/sqb.1998.63.141). Given this, the 'Shiko -35' concept in 3H is a bit awkward as it suggests that sigma70 stops at -10 motifs before planting down on the -35. Considering the cited paper is still in the preprint stages (and did not observe these Shiko -35 emergences), I am concerned about how this particular example will be received by the community. Perhaps more care could be done to verify that this example is consistent with generally accepted mechanisms of promoter recognition or a short clarification could be added to clarify the extent of the analogy.

Thank you for raising this point. We decided to remove the Shiko analogy, because several readers assumed that it relates to the physical binding of RNA polymerase, rather than being an evolutionary mechanism of mutations forming complementary motifs in a stepwise manner.

Lines 323-326: It would be helpful to describe a more systematic approach to defining emergence events into different categories. A clear definition of each category in the methods or main text would help others consistently refer to these concepts in the future. This could be helped by showing the actual parent vs daughter sequences as a supplementary figure to figures 4B, 4D, & 4G.

We agree this could have been more clearly communicated. We have addressed this by (1) simplifying the nomenclatures of these categories and (2) clearly defining these categories, and (3) showing the actual parent vs daughter sequences in Figure 4, and Supplemental Figures S9, S10, S11, and S12. More specifically:

(1) Simplifying the nomenclature. We highlight events where gaining new -10 and -35 boxes can modify the promoter activity of parent sequences with promoter activity. This occurs when a new -10 or -35 box appears that partially overlaps with the -10 or -35 box of the actual promoter. Thus, we rename two terms: hetero-gain and homo-gain, shown in Figure 4B:

(2) We clearly define these categories (lines 430-435):

“We found that these mutations frequently create new boxes overlapping those we had identified as part of a promoter (Fig S9). This occurs when mutations create a -10 box overlapping a -10 box, a -35 box overlapping a 35 box, a -10 box overlapping a -35 box, or a -35 box overlapping a -10 box. We call the resulting event a “homogain” when the new box is of the same type as the one it overlaps, and otherwise a “hetero-gain”. In either case, the creation of the new box does not always destroy the original box.”

In the original manuscript, there was an additional third category, where gaining a -35 box upstream of the promoter’s -35 box, and gaining a -10 box upstream of the promoter’s -10 box decreased expression. We referred to this as a “tandem motif” and it can be found in Figure S12C,D. However, in response to comment “(4) Ignoring or misrepresenting the literature” from Reviewer #3, we carried out an analysis of the binding of H-NS (see Figure 5 and Figure S12). This analysis revealed that this “tandem motif” phenomenon was actually the result of changing the affinity of H-NS to these regions. Thus, the “tandem motif” is probably spurious.

DISCUSSION:Line 378-379: Since hotspots are essentially areas where promoters appear, wouldn't it be obvious that having more hotspots (i.e. areas where more promoters appear) would equate to a higher probability of new promoters? It would be helpful to clarify why this isn't obvious. This could be resolved by adding more complexity to the statement, such as showing that the level of mutual information found in a hotspot or across all hotspots in a sequence is correlated with Pnew.

A fair criticism. In response, we have chosen to remove the analysis of this trend from the manuscript entirely. (Additionally, Pnew and mutual information calculations both relied on the fluorescence scores of daughter sequences, so the finding was circular in its logic.)

Line 394-396: This comparison of findings to Bykov et al should include a bit more justification for the proposed mechanism and how it specifically was observed in this paper. What did they observe and how do these findings relate?

We gladly followed this suggestion, and added the following two paragraphs to the discussion (lines 622-640).

“A previous study randomly mutagenized the appY promoter island upstream of a GFP reporter, and isolated variants with increased and decreased GFP expression. The authors found that variants with higher GFP expression acquired mutations that (1) improve a -10 box to better match its consensus, and simultaneously (2) destroy other -10 and -35 boxes (Bykov et al., 2020). The authors concluded that additional -10 and -35 boxes repress expression driven by promoter islands. Our data challenge this conclusion in several ways.

First, we find that only ~13% of -10 and -35 boxes in promoter islands actually contribute to promoter activity. Extrapolating this percentage to the appY promoter island, ~87% (100% - 13%) of the motifs would not be contributing to its activity. Assuming the appY promoter island is not an outlier, this would insinuate that during random mutagenesis, these inert motifs might have accumulated mutations that do not change fluorescence. Indeed, Bykov et al. (Bykov et al., 2020) also found that a similar frequency of -10 and -35 boxes were destroyed in variants selected for lower GFP expression, which supports this argument. Second, we find no evidence that creating a -10 or -35 box lowers promoter activity in any of our 50 parent sequences. Third, we also find no evidence that destruction of a -10 or -35 box increases promoter activity without plausible alternative explanations, i.e. overlap of the destroyed box with a H-NS site, destruction of the promoter, or simultaneous creation of another motif as a result of the destruction. In sum, -10 and 35 boxes are not likely to repress promoter activity. “

METHODS:Line 500: Could you provide more details on PMR1 (e.g. size, copy number, RBS strength) or a reference? I could not find this easily.

Thank you for pointing out this oversight. In response, we have added the following subsection to the methods (lines 740-748):

“Plasmid MR1 (pMR1)

The plasmid MR1 (pMR1) is a variant of the plasmid RV2 (pRV2) in which the kan resistance gene has been swapped with the cm resistance gene (Guazzaroni and Silva-Rocha, 2014). Plasmid pMR1 encodes the BBa_J34801 ribosomal binding site (RBS, AAAGAGGAGAAA) 6 bp upstream of the start codon for GFP(LVA). The plasmid also encodes a putative RBS (AAGGGAGG) (Cazemier et al., 1999) 5 bp upstream of the start codon for mCherry on the opposite strand.

The plasmid additionally contains the low-to-medium copy number origin of replication p15A (Westmann et al., 2018).

A map of the plasmid is available on the Github repository: https://github.com/tfuqua95/promoter_islands.”

Line 581: What was the sequencing instrument &/or depth?

We now report this information as follows (Methods, lines 918-922):

“Illumina sequencing

The amplicon pool was sequenced by Eurofins Genomics (Eurofins GmbH, Germany) using a NovaSeq 6000 (Illumina, USA) sequencer, with an S4 flow cell, and a PE150 (Paired-end 150 bp) run. In total, 282’843’000 reads and 84’852’900’000 bases were sequenced. Raw sequencing reads can be found here: https://www.ncbi.nlm.nih.gov/bioproject/1071572.”

SUPPLEMENT:Supplementary Figure 2: Why does the GFP control produce a bimodal distribution?

The GFP+ culture was inoculated directly from a glycerol stock. The bimodal distribution probably results from a subset of the bacteria having lost the GFP-coding insert, because the left-most peak coincides with the negative control.

**Reviewer #2 (Recommendations For The Authors):**
This paper would benefit from a clear definition of what constitutes an active promoter as this is only mentioned as justification for the use of arbitrary values for fluorescence.

Good point. To clarify, we now include this new paragraph in the introduction (lines 112-119):

“In this study, we define a promoter as a DNA sequence that drives the expression of a (fluorescent) protein whose expression level, measured by its fluorescence, is greater than a defined threshold. We use a threshold of 1.5 arbitrary units (a.u.) of fluorescence. This definition does not distinguish between transcription and translation. We chose it because protein expression is usually more important than RNA expression whenever natural selection acts on gene expression, because it is the primary phenotype visible to natural selection (Jiang et al., 2023).”

There needs to be a clear distinction in the use of the word sequences as often interchange sequences when meaning the 25 parent sequences and then the 50 possible sequences directions the promoter can act. It is confusing going from one to the other.

We agree that this distinction is important. To make it clearer, we now introduce an additional term (lines 119-130). Our experiments start from 25 promoter island fragments (P1-P25), which we now call template sequences. Each template sequence comprises both DNA strands. The parent sequences are the top and bottom strands of each template sequence. Therefore, there are now 50 parent sequences (P1-GFP, P1-RFP, P2-GFP…, P25-RFP). By treating each strand as its own sequence, we no longer have to refer to the strand, avoiding the earlier confusion.

The description of the hotspots is often unclear and trying to determine if 3 out of 9 hotspots come from one parent sequence or multiple is not possible. A table denoting this information would be most helpful.

We agree, and now provide this information in Data S3.

Finally, the description of the proposed mechanism of promoter activation via mutation of motifs should not be in the results but in the discussion, as it has insufficient evidence and would require further experimental validation.

We remedied this problem by providing experimental validation of the proposed mechanisms. Specifically, we created the precise mutations that caused a loss or gain of a -10 or a -35 box, and measured the level of gene expression they drive with a plate reader. Because we chose to provide this experimental validation, we opted to leave the mechanisms of promoter activation in the results section.

The (Fuqua and Wagner 20023) paper is not in the references.

We have added Fuqua and Wagner, BiorXiv 2023 to the references.

I enjoyed the paper and wish the authors the best for their future work.

Thank you for taking the time to review our manuscript!

**Reviewer #3 (Recommendations For The Authors):**
The paper has major flaws. For example:The data need to be analysed with correct promoter sequence element sequences (TTGACA for the -35 element).

The discrepancy lies in the frequency of A’s vs C’s at position #5 of the PWM. Our PWM was built with more A’s than C’s at this position, but also includes C’s in this position. However, we respectfully disagree that using a different -35 box PWM is going to change the outcomes of our study. First, positions 4-6 of the PWM barely have any information content (bits) compared to positions 1-3 (see Fig 1A). This assertion is not just based on our own PWM, but based on ample precedent in the literature. In PMID 14529615, TTG is present in 38% of all -35 boxes, but ACA only 8%. In PMID 29388765, with the -10 instance TATAAT, the -35 instance TTGCAA yields stronger promoters compared to the -35 instance TTGACA (See their Figure 3B). In PMID 29745856 (Figure 2), the most information content lies in positions 1-3, with the A and C at position 5 both nearly equally represented, as in our PWM. In PMID 33958766 (Figure 1) an experimentally-derived -35 box is even reduced to a “partial” -35 box which only includes positions 1 and 2, with consensus: TTnnnn. Additionally, the -35 box PWM that we used significantly and strongly correlates with an experimentally derived -35 box (see Supporting Information from Figure S4 of Belliveau et al., PNAS 2017. Pearson correlation coefficient = 0.89). We now provide DNA sequences for each of the figures to improve accessibility and reproducibility. A reader can now use any PWM or method they wish to interpret the data.

The data need to be analysed taking into account the role of other promoter elements and sequences for translation.

Point well taken.

Thank you for bringing this oversight to our attention. We have performed two independent analyses to explore the role of TGn in promoter emergence in evolution. First, we computationally searched for -10 boxes with the bases TGn immediately upstream of them in the parent sequences, and found 18 of these “extended -10 boxes” in the parents (lines 143145):

“On average, each parent sequence contains ~5.32 -10 boxes and ~7.04 -35 boxes (Fig S1). 18 of these -10 boxes also include the TGn motif upstream of the hexamer.”

However, only 20% of these boxes were found in parents with promoter activity (lines 182-185):

“We also note that 30% (15/50) of parents have the TGn motif upstream of a -10 box, but only 20% (3/15) of these parents have promoter activity (underlined with promoter activity: P4-RFP, P6-RFP, P8-RFP, P9-RFP, P10-RFP, P11GFP, P12-GFP, P17-GFP, P18-GFP, P18-RFP, P19-RFP, P22-RFP, P24-GFP, P25-GFP, P25-RFP).”

Second, we computationally searched through all of the daughter sequences to identify new -10 boxes with TGn immediately upstream. We found 114 -10 boxes with the bases TGn upstream. However, only 5 new -10 boxes (2 with TGn) were associated with increasing fluorescence (lines 338-345):

“Mutations indeed created many new -10 and -35 boxes in our daughter sequences. On average, 39.5 and 39.4 new 10 and -35 boxes emerged at unique positions within the daughter sequences of each mutagenized parent (Fig 3A,B), with 1’562 and 1’576 new locations for -10 boxes and -35 boxes, respectively. ~22% (684/3’138) of these new boxes are spaced 15-20 bp away from their cognate box, and ~7.3% (114/1’562) of the new -10 boxes have the TGn motif upstream of them. However, only a mere five of the new -10 boxes and four of the new -35 boxes are significantly associated with increasing fluorescence by more than +0.5 a.u. (Fig 3C,D).”

In addition, we now study the role of UP elements. This analysis showed that the UP element plays a negligible role in promoter emergence within our dataset. It is discussed in a new subsection of the results (lines 591-608).

“The UP-element does not strongly influence promoter activity in our dataset.

The UP element is an additional AT-rich promoter motif that can lie stream of a -35 box in a promoter sequence (Estrem et al., 1998; Ross et al., 1993). We asked whether the creation of UP-elements also creates or modulates promoter activity in our dataset. To this end, we first identified a previously characterized position-weight matrix for the UP element (NNAAAWWTWTTTTNNWAAASYM, PWM threshold score = 19.2 bits) (Estrem et al., 1998) (Fig S13A). We then computationally searched for UP-element-specific hotspots within the parent sequences, i.e., locations in which mutations that gain or lose UP-elements lead to significant fluorescence increases (Mann-Whitney U-test, Fig S7 and methods. See Data S8 for the coordinates, fluorescence changes, and significance). The analysis did not identify any UP elements whose mutation significantly changes fluorescence.

We then repeated the analysis with a less stringent PWM threshold of 4.8 bits (1/4th of the PWM threshold score). This time, we identified 74 “UP-like” elements that are created or destroyed at unique positions within the parents. 23 of these motifs significantly change fluorescence when created or destroyed. However, even with this liberal threshold, none of these UP-like elements increase fluorescence by more than 0.5 a.u. when gained, or decrease fluorescence by more than 0.5 a.u. when lost (Fig S13B). This finding ultimately suggests that the UP element plays a negligible role in promoter emergence within our dataset.”

Collectively, these additional analyses suggest that the presence of TGn plus a -10 box is insufficient to create promoter activity, and that the UP element does not play a significant role in promoter emergence or evolution.

The full sequences used need to be provided and mutations resulting in new promoters need to be shown.

To Figures 3, 4, 5, and Supplemental Figures S8, S9, S10, S11, and S12, we have added the sequences which created or the destroyed the promoters, and their PWM scores.

The paper needs to be rewritten to take into account the relevant literature on (i) promoter islands (i.e. sections of horizontally acquired AT-rich DNA) (ii) generation and loss of promoters by mutation.

We have rewritten the introduction. The majority of these points are now addressed in the following two new paragraphs (lines 92-112):

“Recent work shows that mutations can help new promoters to emerge from promoter motifs or from sequences adjacent to such motifs (Bykov et al., 2020; Fuqua and Wagner, 2023; Yona et al., 2018). However, encoding -10 and -35 boxes is insufficient to drive complete transcription of a gene coding sequence. For instance, the *E. coli* genome contains clusters of -10 and -35 boxes that are bound by RNA polymerase and produce short oligonucleotide fragments, but rarely create complete transcripts. Such clusters are called promoter islands, and are strongly associated with horizontally-transferred DNA (Bykov et al., 2020; Panyukov and Ozoline, 2013; Purtov et al., 2014; Shavkunov et al., 2009).

There are two proposed explanations for why promoter islands do not create full transcripts. First, the TF H-NS may repress promoter activity in promoter islands. This is because in a Δhns background, transcript levels from the promoter islands increases (Purtov et al., 2014). However, mutagenizing a specific promoter island (appY) until it transcribes a GFP reporter, reveals that in-vitro H-NS binding does not significantly change when GFP levels increase (Bykov et al., 2020). Thus, it is not clear whether H-NS actually represses the complete transcription of these sequences. The second proposed explanation is that excessive promoter motifs silence transcription. The aforementioned study found that promoter activity increases when mutations improve a -10 box to better match its consensus (TAAAAAT→TATACT), while simultaneously destroying surrounding -10 and -35 boxes (Bykov et al., 2020). However, we note that if these surrounding motifs never contributed to GFP fluorescence to begin with, then mutations could also simply have accumulated in them during random mutagenesis without affecting promoter activity.

In closing, we would like to thank all three reviewers again for your time to engage with this manuscript.

Summary of specific changes that we have made to each section of the manuscript

• Abstract

- We updated the abstract to include the finding that more than 1’500 new -10s and 35s are created in our dataset, but only ~0.3% of them actually create de-novo promoter activity.

- We no longer highlight the conclusion that the majority of promoters emerge and evolve from -10 and -35 boxes.

• Introduction

- We have added more background information about the UP-element and the TGn motif.

- We better describe the promoter islands and the results identified by Bykov et al., 2020.

• Results: Promoter island sequences are enriched with motifs for -10 and -35 boxes.

- We clarify how the -10 and -35 PWMs we use were derived.

- We refer to the 25 promoter island fragments as “Template sequences” (P1-P25). The “parent sequences” now correspond to the top and bottom strands of each template (N=50, P1-GFP, P1-RFP, P2-GFP, …, P25-RFP).

- We elaborate that ~7% of the -10 boxes in the template sequences have the TGn motif.

- In the previous version of the manuscript, if there were overlapping -10 boxes or overlapping -35 box, we counted these to be a single -10 box or a single -35 box, respectively. In the new version of the manuscript, we now treat each motif as an independent box. Because of this, the number of -10 and -35 boxes per parent have slightly increased.

•Results: Non-promoters vary widely in their potential to become promoters.

- We make a clear distinction between promoters and non-promoters, and define the parent sequences.

- We note that only 20% of parents with an “extended -10 box” have promoter activity.

• Results: Promoter emergence correlates with minute differences in background promoter levels.

- We added an analysis where we compare Pnew to the parent fluorescence levels, even if they are below 1.5 a.u. We find that the distribution of Pnew matches a sigmoid function.

• Results: Promoter emergence does not correlate with simple sequence features

- We added an analysis comparing k-mer counts to Pnew.

- We updated the way we count -10 and -35 boxes, and recalculated the correlation with Pnew. The P and R2 values have changed, but Pnew still does not significantly correlate with -10 or -35 box counts.

• Results: Promoters emerge and evolve only from specific subsets of -10 and -35 boxes

- We have added an analysis where we computationally scramble the wild-type parent sequences while maintaining the coordinates of the mutual information hotspots. This reveals that the overlap with -10 and -35 motifs is not a coincidence of dense promoter motif encoding.

We found a computational error in our analysis and updated the percent overlap between -10 boxes and -35 boxes with mutual information hotspots. The results are similar. o 14% of -10 boxes overlap with hotspots with our new way of defining -10 and -35 boxes.

• Results: New -10 and -35 boxes readily emerge, but rarely lead to de-novo promoter activity

- We quantify how often a new -10 and -35 box is created at a unique position within our collection of promoter fragments, and how often this results in a -10 and -35 box being appropriately spaced, and how often this actually leads to de-novo promoter activity. o We quantify how often a TGn sequence lies upstream of a new -10 box.

• Results: Promoters can emerge when mutations create motifs but not by destroying them.

- For each example, we added the DNA sequences of the wild-type region of interest and the mutant region of interest that results in the gain of promoter activity, and their respective PWM scores.

- We created constructs to validate each example by testing their fluorescence on a plate reader.

- We removed the P1-GFP example from the main figure, as it was a false-positive in the dataset. It is now in Fig S8.

- We removed the Shiko Emergence metaphor because it could be confused with a binding mechanism for RNA polymerase.

• Results – Gaining new motifs over existing motifs increases and decreases promoter activity.

- We removed the “Tandem motif” because it is more likely caused by H-NS binding.

- We renamed the mechanisms to be “hetero-gain” and “homo-gain” for simplicity, and clearly define how we classified each sequence into each category.

- We now include the DNA sequences, the PWM scores, the spacer lengths, and the fluorescence values from constructs harboring the predicted point mutations.

• Results – Histone-like nucleoid-structuring protein (H-NS) represses P12-RFP and P22-GFP.

- This is a new analysis, which explores the role of the TF H-NS in repressing the parent sequences.

- We identified putative H-NS motifs in P12-RFP and P22-GFP.

- We show experimentally that in a H-NS null background, a bidirectional promoter (P20) becomes unidirectional, even though P20 does not contain an obvious H-NS motif.

- In the original version of the manuscript, we describe a phenomenon where gaining a -35 box upstream of a promoter’s -35 box, or a -10 box upstream of a promoter’s -10 box significantly decreases expression. We called this phenomenon a “tandem motif.” However, in the newest version of the manuscript, we find that these fluorescence decreases are rescued in a H-NS null background, suggesting the finding was actually due to H-NS binding modulation and not -10 and -35 boxes.

• Results – The UP-element does not strongly influence promoter activity in our dataset.

We used a PWM for the UP element to see if gaining or losing UP motifs was significantly correlated with increasing or decreasing expression. Even with a liberal PWM threshold, the analysis did not find any UP elements.

• Discussion

- We rewrote the discussion to account for the new analyses and the results on H-NS, the UP-element, and the extended -10.

- We better explain how our results clash with the results from the Bykov paper.

- We fit our results into the context of David Grainger’s papers.

• Methods

- Added an explanation about pMR1.

- Added methods describing how we created the point mutation constructs.

- Added the methods for the plate reader.

- Added the methods for Illumina sequencing.

- Added the methods for the sigmoid curve-fitting.

• Figure 1

- Panel E compares how Pnew (the probability of a daughter sequence having a fluorescence score greater than 1.5 a.u.) associates with the fluorescence scores of each parent sequence.

- Panel F was originally in Figure S5. In the originally submitted version of the manuscript, if there were overlapping -10s or overlapping -35s, we counted these to be a single -10 or a single -35, respectively. In the new version of the manuscript, we now treat each motif as an independent box. Because of this, the r2 and p values have changed, but the conclusions have not (Pnew still does not significantly correlate with -10 or -35 box counts).

• Figure 2

- Panel C now includes a stacked barplot showing the percentage of -10 and -35 boxes that overlap with mutual information hotspots when the parent sequences are randomly scrambled computationally.

• Figure 3

- Panels A-C were added to explain how we define a new -10/-35 box, how many such new boxes each parent has. These panels also illustrate how we associate the presence or absence of a motif with significant changes in fluorescence scores of the daughter sequences.

- We moved the example of P1-GFP to Figure S8 because when we tested the specific mutation which leads to gaining the -10 box, fluorescence did not change.

- We now include the DNA sequences, the PWM scores, the spacer lengths, and the fluorescence values from reporter constructs harboring the point mutations predicted by our computational analyses.

- Cartoons of RNA polymerase have been removed.

• Figure 4

- The tandem-motif has been removed from the figure.

- Cartoons of RNA polymerase have been removed.

- We now include the DNA sequences, the PWM scores, the spacer lengths, and the fluorescence values from constructs harboring the point mutations predicted by our computational analyses.

• Figure 5

- This is a new figure analyzing the role of H-NS in promoter evolution and emergence.

• Figure S4

- Panel B now shows the wild-type parent scores and their standard deviations from the sort-seq experiment.

• Figure S5

- Panels with -10 and -35 box counts moved to Figure 1.

- The panel comparing Pnew to hotspot counts was removed.

- Correlations between different k-mers and Pnew are added to panels C-H.

• Figure S8

- We now include the DNA sequences, the PWM scores, the spacer lengths, and the fluorescence values from constructs harboring the point mutations predicted by our computational analyses.

• Figure S9

- We now include the DNA sequences, the PWM scores, the spacer lengths, and the fluorescence values from constructs harboring the point mutations predicted by our computational analyses.

• Figure S10

- We now include the DNA sequences, the PWM scores, the spacer lengths, and the fluorescence values from constructs harboring the point mutations predicted by our computational analyses.

• Figure S11

- Added DNA sequences and PWM scores.

• Figure S12

- A new figure with further insights about H-NS.

• Figure S13

- A new figure regarding the UP-element analysis.

• Figure S14

- Added Panel D to show how we created mutant reporter constructs for validation.